# Large depth differences between target and flankers can increase crowding: Evidence from a multi-depth plane display

**Samuel P Smithers\*, Yulong Shao, James Altham, Peter J Bex\***

Department of Psychology, Northeastern University, Boston, United States

**Abstract** Crowding occurs when the presence of nearby features causes highly visible objects to become unrecognizable. Although crowding has implications for many everyday tasks and the tremendous amounts of research reflect its importance, surprisingly little is known about how depth affects crowding. Most available studies show that stereoscopic disparity reduces crowding, indicating that crowding may be relatively unimportant in three-dimensional environments. However, most previous studies tested only small stereoscopic differences in depth in which disparity, defocus blur, and accommodation are inconsistent with the real world. Using a novel multi-depth plane display, this study investigated how large (0.54–2.25 diopters), real differences in target-flanker depth, representative of those experienced between many objects in the real world, affect crowding. Our findings show that large differences in target-flanker depth increased crowding in the majority of observers, contrary to previous work showing reduced crowding in the presence of small depth differences. Furthermore, when the target was at fixation depth, crowding was generally more pronounced when the flankers were behind the target as opposed to in front of it. However, when the flankers were at fixation depth, crowding was generally more pronounced when the target was behind the flankers. These findings suggest that crowding from clutter outside the limits of binocular fusion can still have a significant impact on object recognition and visual perception in the peripheral field.

**\*For correspondence:**
s.smithers@northeastern.edu
(SPS);
p.bex@northeastern.edu (PJB)

**Competing interest:** The authors declare that no competing interests exist.

## Editor's evaluation

Using a novel multi-depth plane display, this important study reveals that crowding decreases with small depth differences between the target and flankers but increases with larger depth differences. The evidence supporting the claims is convincing, although the explanation of some findings is somewhat speculative. This paper will be of interest to visual scientists and neuroscientists.

## Introduction

Visual crowding is a phenomenon in which a peripherally viewed object, that is easily identifiable in isolation, becomes harder to recognize or identify when it is surrounded by other objects, despite still being clearly visible (*Bouma, 1970*; *Herzog et al., 2015*; *Levi, 2008*; *Pelli, 2008*; *Whitney and Levi, 2011*). The importance of crowding is reflected by the tremendous amount of research on the topic and it has been extensively reviewed over the years (e.g. *Coates et al., 2021*; *Herzog et al., 2015*; *Manassi and Whitney, 2018*; *Pelli, 2008*; *Strasburger et al., 2011*; *Whitney and Levi, 2011*). This high level of interest is not surprising given that the natural environment is usually cluttered and mostly viewed in the peripheral visual field. Consequently, it is assumed that crowding has a large impact on

**eLife digest** While human eyesight is clearest at the point where the gaze is focused, peripheral vision makes objects to the side visible. This ability to detect movement and objects in a wider field of vision helps people to have a greater awareness of their surroundings. However, it is more difficult to identify an object using peripheral vision when it is surrounded by other items. This phenomenon is known as crowding and can affect many aspects of daily life, such as driving or spotting a friend in a crowd.

In our three-dimensional world, peripheral objects are often at different distances. This variation in depth could influence the effect of crowding, yet little is known about its effect. While previous research has investigated the effect of small differences in depth on crowding, the studies did not replicate real-world conditions.

To replicate depths that are likely to be encountered in the real world, Smithers et al. created a display using multiple screens positioned 0.4, 1.26 and 4 meters from the viewer. Images were displayed on the screens and researchers measured how well study participants could identify a target image when it was surrounded by similar, nearby images displayed closer or further away than the target. The experiments showed that most viewers are less able to recognize a target object when there are surrounding items and this effect is worsened when the items are separated from the object by large differences in depth.

The findings show that instead of diminishing the effect of crowding – as suggested by previous studies with small depth differences – large depth differences that more closely recreate those encountered in the real world can amplify the effect of crowding. This greater understanding of how humans process objects in three-dimensional environments could help to better estimate the impact of crowding on people with eye and neurological disorders. In turn, the information could be used to design environments that are easier for such individuals to navigate.

our daily lives with implications for many everyday visual tasks including driving (*Xia et al., 2020*), reading (*Legge, 2007*), and face recognition (*Kalpadakis-Smith et al., 2018*; *Louie et al., 2007*). Crowding also has important clinical implications for people with glaucoma (*Ogata et al., 2019*; *Shamsi et al., 2022*), as well as for other disorders including central vision impairment (e.g. macular degeneration), amblyopia and dyslexia (*Whitney and Levi, 2011*). Through this body of research, we have been able to build a good understanding of the factors that mediate and moderate crowding and develop unifying theories (e.g. *Balas et al., 2009*; *Harrison and Bex, 2015*; *Keshvari and Rosenholtz, 2016*; *van den Berg et al., 2012*; *van den Berg et al., 2010*). However, despite the large body of literature, one area that remains surprisingly understudied is how depth affects crowding. This is important because we live in a three-dimensional environment and so to understand the significance of crowding in the real world it is essential to understand the role of depth information.

*Kooi et al., 1994* were the first to investigate how depth influences crowding. The authors found that compared to when a peripheral target and flankers were at the same depth, there was a release from crowding when the target and flankers were stereoscopically presented in crossed (front) and uncrossed (behind) disparities (*Kooi et al., 1994*). *Sayim et al., 2008* found that when flankers were at fixation depth, there was less foveal crowding when the target was stereoscopically presented in front or behind the flankers. *Felisberti et al., 2005* also found an effect of depth on crowding in the parafovea, but to a more limited extent. They reported a release from crowding in two out of three subjects when the target appeared behind the flankers, but only in one of the three subjects when the target was in front of the flankers (*Felisberti et al., 2005*). *Astle et al., 2014* extended on previous research by testing the effect of depth on crowding over a range of stereoscopically produced target-flanker disparities (±800 arcsec). In line with previous studies, *Astle et al., 2014* found that when the target was at fixation depth, increasing target-flanker disparity systematically reduced crowding. They also found that the release from crowding was asymmetric whereby crowding was greater when the flankers were in front of the target compared to behind it (*Astle et al., 2014*).

Taken together these studies appear to paint the picture that a difference in depth between a target and flankers reduces crowding. This could challenge the concept that crowding is ubiquitous in three-dimensional natural environments (*Whitney and Levi, 2011*) because it implies that

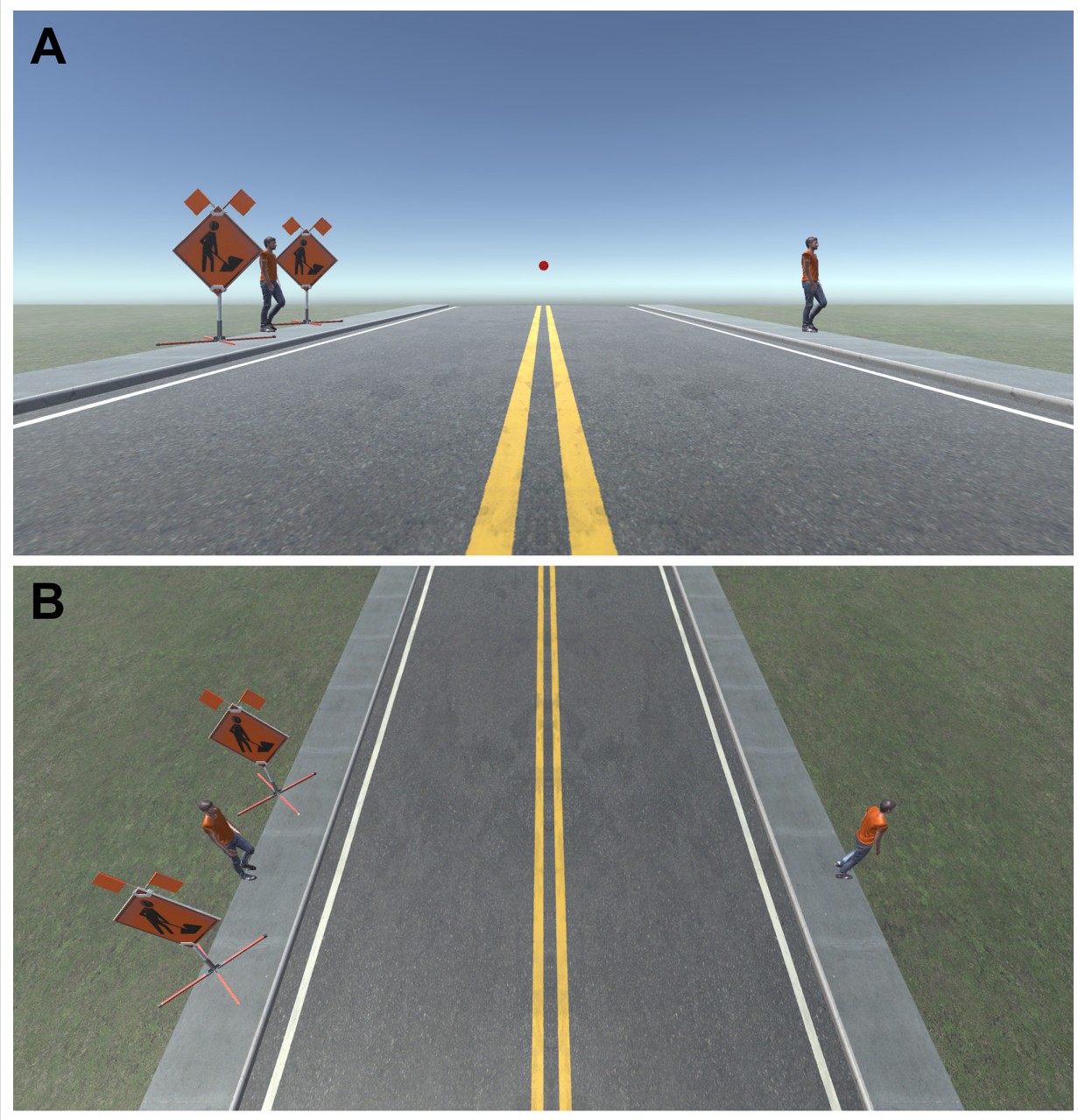

**Figure 1.** Example of crowding in a simulated everyday scene. (**A**) When fixating on the red dot above the horizon, it is relatively easy to spot and recognize the person crossing on the right. However, it is more difficult, or impossible, to recognize the person about to cross the road on the left because of the presence of nearby signs. Images like this one are often used to illustrate how crowding is ubiquitous in everyday natural scenes. (**B**) However, in real life the two signs and the person on the left may be several meters apart and so each would appear at a different depth from the observer. Currently, little is known about how large, real differences in depth between objects affect crowding. See acknowledgments for image attributions.

depth differences between natural objects could prevent crowding from being a problem (*Figure 1*). However, all the above studies tested only small depth differences between target and flankers that do not reflect the often-large differences in depth between objects in the real world. Furthermore, the generalizability of these studies to real world depth is also limited by the fact that depth cues were generated stereoscopically. One potential problem with stereo displays is that disparity information often does not match with accommodation or defocus blur which can detrimentally affect depth perception (*Hoffman et al., 2008*; *Watt et al., 2005*).

To our knowledge, four studies have investigated the effect of depth on crowding using a real depth display. All four studies used two orthogonally positioned screens viewed through a half-transparent mirror allowing stimuli to be displayed on two different depth planes simultaneously (*Eberhardt et al., 2021*; *Eberhardt and Huckauf, 2019*; *Eberhardt and Huckauf, 2017*; *Eberhardt and Huckauf, 2020*). *Eberhardt and Huckauf, 2017* kept a target and flankers at the same depth and presented them 0.06 diopters (i.e. 1/focal length in meters), in front, behind or at the same depth as fixation. The authors found that this small difference in depth between the stimuli and fixation planes had no effect on the amount of crowding compared to when everything was presented at the fixation plane (*Eberhardt and Huckauf, 2017*). *Eberhardt and Huckauf, 2019* repeated and expanded on this work with the addition of a ±0.1 diopter depth difference between fixation and the stimuli in addition to the ±0.06 diopter condition. The results showed that crowding was greater when the stimuli were furthest from fixation compared to when they were closer to fixation, regardless of whether the stimuli were in front or behind the fixation plane (*Eberhardt and Huckauf, 2019*). More recently, *Eberhardt and Huckauf, 2020* used the same approach to investigate the effect of depth on crowding when the target and flankers were separated in depth. When flanker depth was varied while the target was at fixation depth, crowding increased with increasing target-flanker depth difference (*Eberhardt and Huckauf, 2020*). This result is the opposite to the findings of other studies in which small differences in depth between the target and flankers were introduced with stereoscopic disparity (*Astle et al., 2014*; *Felisberti et al., 2005*; *Kooi et al., 1994*; *Sayim et al., 2008*). Furthermore, the authors also found that when flanker depth was varied while the target was at fixation depth, crowding was greater when the flankers were behind the target and fixation compared to in front of it (*Eberhardt and Huckauf, 2020*). When target depth was varied while the flankers were at

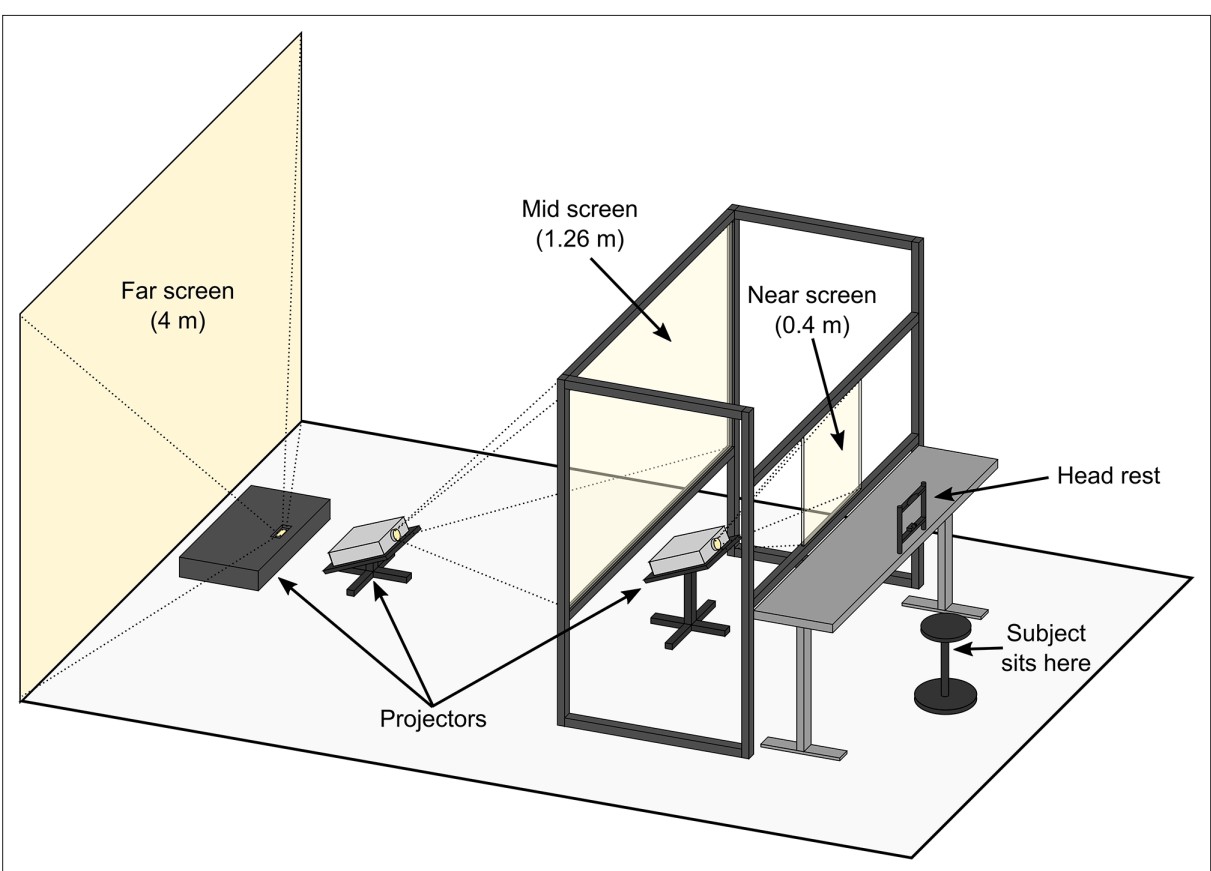

**Figure 2.** Illustration of the multi-depth plane display setup used for this study. The near and mid screens consisted of transparent acrylic covered with ClearBright display film (see Methods for details) that enabled projected images to be displayed on to the screens while preserving their transparency (*Hsu et al., 2014*). This enabled observers to simultaneously view adjacent stimuli presented at different depth planes at the same time. The screens were temporally synchronized and spatially aligned (see Methods) to ensure display timing and position were consistent across the three depth planes.

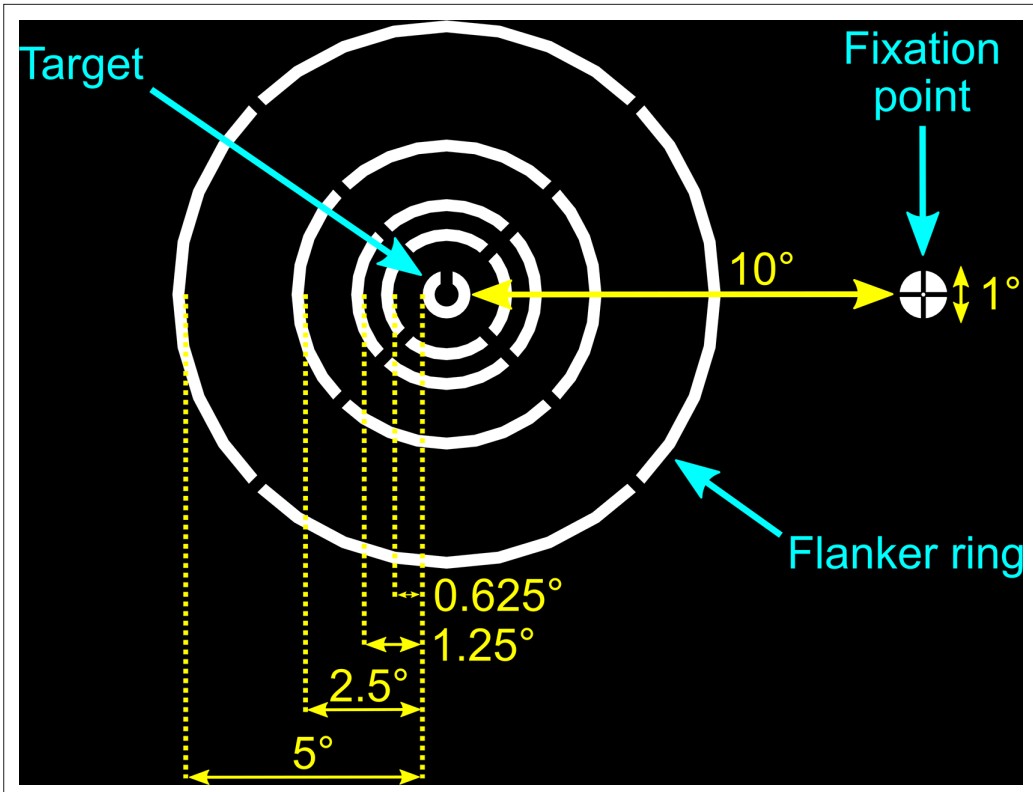

**Figure 3.** Schematic illustration of experimental stimuli and fixation marker. In all conditions, except the no-flanker condition, the randomly orientated target (a white Landolt-C) was surrounded by a white flanker ring at one of four possible target-flanker distances. The stimuli were viewed against a black background.

fixation, there was more crowding when the target was in front of the flankers and fixation compared to when it was behind them (*Eberhardt and Huckauf, 2020*).

In light of the above discrepancies, and the prevalence of depth information in real world scenes, there is a pressing need for a better understanding of how visual crowding is affected by depth differences representative of those experienced between objects in the real world. Therefore, using a novel multi-depth plane display consisting of screens at 0.4 m (near screen), 1.26 m (mid screen), and 4 m (far screen) viewing distances (*Figure 2*), we investigate how crowding is affected by large, real differences in target-flanker depth. These viewing distances were chosen because 4 m is close to optical infinity while 0.4 m is the clinical standard for near distance, with 1.26 m being in the middle of these two extremes in log units. Given in diopters, the near, mid, and far screens were at 2.5 diopters, 0.79 diopters and 0.25 diopters receptively.

## Results

For the main study, naive observers were tasked with reporting the orientation of a peripherally viewed target (*Figure 3*) in a series of five experiments (see Methods for details and *Figure 4* for example trial). The target was surrounded by a flanker ring with a target-flanker spacing of 5°, 2.5°, 1.25°, 0.625°, or infinity, i.e., no flanker ring (*Figure 3*). In Experiments 1 and 2, the target was always presented at fixation depth while the depth of the flanker ring was varied. Conversely, in Experiments 3 and 4 the flanker ring was presented at fixation depth while target depth was varied. Finally, for Experiment 5, the target, flanker ring and fixation were presented together on the same screen at three different depths. To quantify crowding for each stimulus condition we used the reported target orientation across repeated trials to calculate perceptual error (see Methods), which is reported in degrees.

Due to the nature of the experimental set up, and the large distances being tested, it is possible that head movements, diplopia, or local suppression (see Discussion) could have resulted in differences in

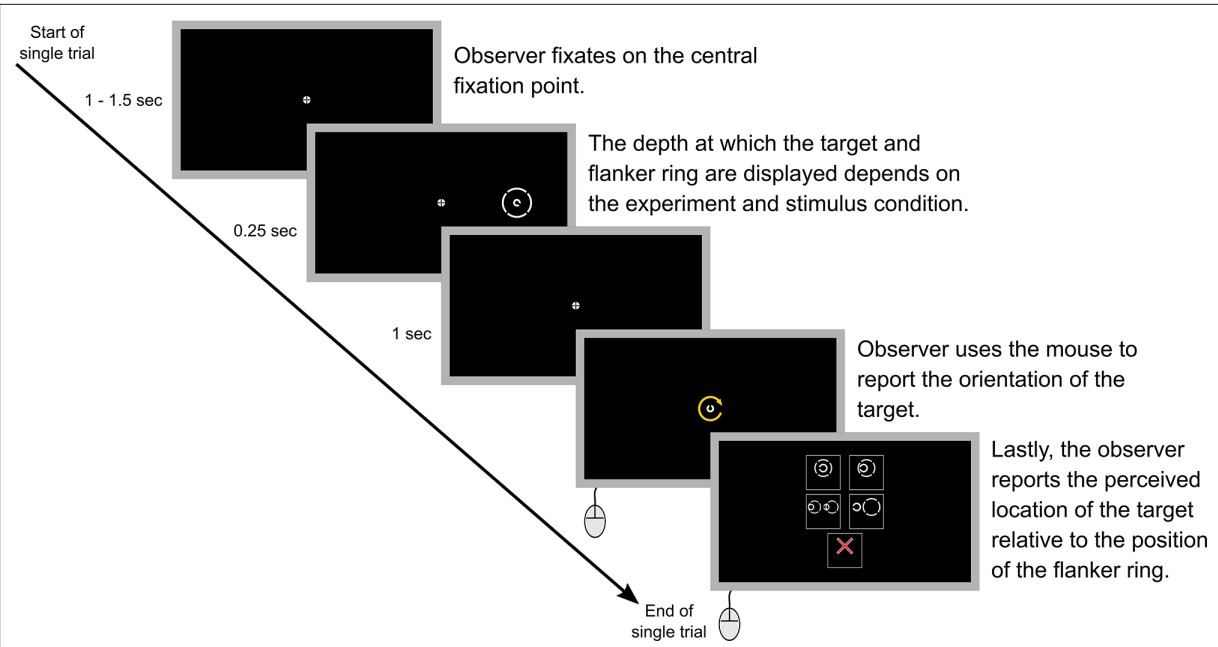

**Figure 4.** A step-by-step example of a single trial. Note that in the actual experiments a text description of each of the five options in step five was also included within the option boxes. The text descriptions for the different options were: 'Target in center of ring' (top left), 'Target inside ring but NOT in the center' (top right), 'Ring obstructs target' (middle left), 'Target outside ring' (middle right), and 'Unsure or no ring' (bottom center). Step five was not included in Experiment 5 as all parts of the stimulus were always presented at the same depth.

the perceived position of the target relative to the flanker ring. Therefore, in all experiments except Experiment 5, observers were also asked to report the perceived location of the target relative to the position of the flanker ring (*Figure 4*). This enabled us to examine how perceptual error was affected by factors such as target misalignment and possible diplopia (see Discussion).

## Perceptual error
### Target always at fixation depth with variable flanker depth
#### Experiment 1: Target and fixation at 1.26 m with flanker ring presented at each depth (Figure 5A and Figure 5—figure supplement 1A)
Perceptual error was higher when the flanker ring was displayed in front of, or behind, the target and fixation plane (mixed effects multiple linear regression models compared via LRT for effect of flanker depth: $\chi^2_{(2)}$=20.67, p<0.001). As expected, perceptual error decreased as target-flanker spacing increased (effect of target-flanker spacing: $\chi^2_{(1)}$=31.71, p<0.001). There was no significant interaction between flanker depth and target-flanker spacing ($\chi^2_{(2)}$=5.56, p=0.062).

#### Experiment 2: Target and fixation at 0.4 m or 4 m with flanker ring presented at each depth (Figure 5B and Figure 5—figure supplement 1B)
Perceptual error tended to increase with increasing flanker depth when the flanker ring was behind the target and fixation plane, however, the effect of flanker depth was much weaker when the flanker ring was in front of the target and fixation (interaction between target-fixation depth and flanker depth: $\chi^2_{(2)}$=17.18, p<0.001). Perceptual error decreased with increasing target-flanker spacing, in a manner corresponding to target-fixation depth (interaction between target-flanker spacing and target-fixation depth: $\chi^2_{(1)}$=9.2, p=0.002). There was no significant interaction between target-flanker spacing and flanker screen and ($\chi^2_{(2)}$=4.52, p=0.104).

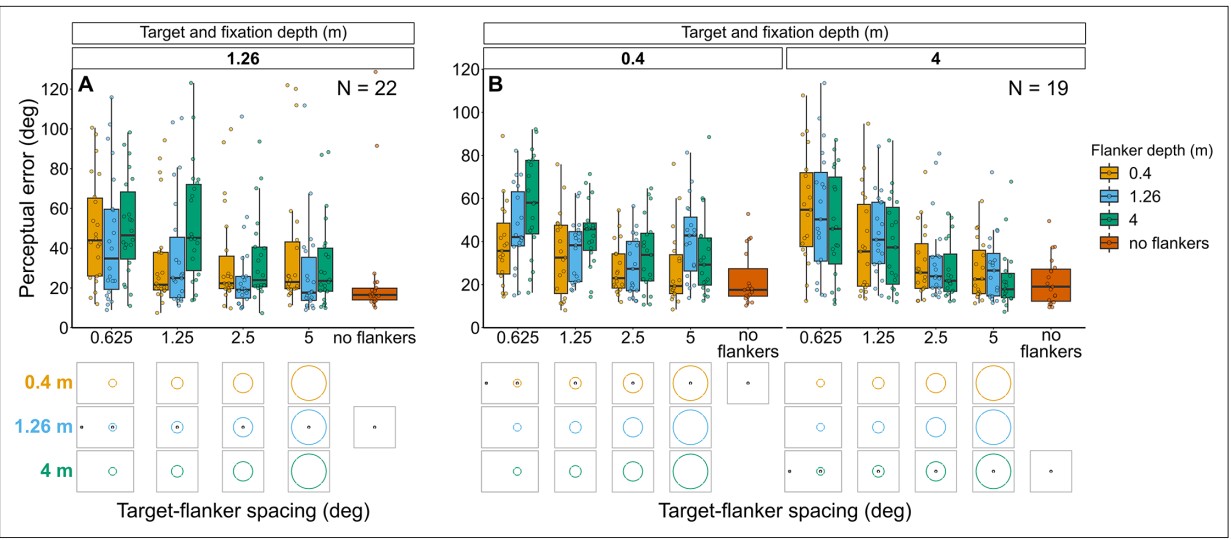

**Figure 5.** Perceptual error results for Experiments 1 (**A**) and 2 (**B**) in which flanker depth was varied while the target was always at fixation depth. Box plots show medians plus the interquartile range (IQR), whiskers are the lowest and highest values that are within 1.5 times the IQR from the upper and lower quartiles. Points show the perceptual error for each individual observer. The visual aid below each graph shows the depth of the flanker ring (based on the same color key as the box plot) relative to the depth of the fixation point and target (shown in black) for each stimulus condition. The relative sizes of, and distances between, the fixation point, target, and flanker ring are to scale. Note that the fixation point is only shown for the 0.625° target-flanker spacing due to limited space within the figure.

The online version of this article includes the following figure supplement(s) for figure 5:

**Figure supplement 1.** Perceptual error for Experiments 1 (**A**) and 2 (**B**) calculated only from trials in which the observer reported seeing the target inside the flanker ring.

### Flanker ring always at fixation depth with variable target depth

#### Experiment 3: Flanker ring and fixation at 1.26 m with target presented at each depth (Figure 6A and Figure 6—figure supplement 1A)

Overall, perceptual error was higher when the target was displayed in front of, or behind, the flanker ring and fixation (effect of target depth: $\chi^2_{(2)}=36.93$, p<0.001), with the greatest perceptual error occurring when the target was behind the flanker ring and fixation. Perceptual error decreased as target-flanker spacing increased (effect of target-flanker spacing: $\chi^2_{(1)}=45.99$, p<0.001). There was no significant interaction between target depth and target-flanker spacing ($\chi^2_{(2)}=3.55$, p=0.17).

#### Experiment 4: Flanker ring and fixation at 0.4 m or 4 m with target presented at each depth (Figure 6B and Figure 6—figure supplement 1B)

Perceptual error was greatest when the target was displayed behind the flanker ring and fixation, while the effect of depth was less prominent when the target was displayed in front of the flanker ring and fixation (interaction between flanker-fixation depth and target depth: $\chi^2_{(2)}=61.87$, p<0.001). Perceptual error decreased as target-flanker spacing increased (effect of target-flanker spacing: $\chi^2_{(1)}=29.92$, p<0.001). There was no significant interaction between flanker-fixation depth and target-flanker spacing ($\chi^2_{(1)}=0.14$, p=0.71) nor between target depth and target-flanker spacing ($\chi^2_{(2)}=1.43$, p=0.488).

### Target and flanker ring at fixation depth

#### Experiment 5: Target and flanker ring always at fixation depth (Figure 7)

Perceptual error decreased as target-flanker spacing increased (effect of target-flanker spacing: $\chi^2_{(1)}=64.8$, p<0.001). There was also a significant effect of target-flanker-fixation depth ($\chi^2_{(2)}=8.75$, p=0.013) resulting from a higher perceptual error for stimuli displayed at 4 m. There was no significant interaction between target-flanker-fixation depth and target-flanker spacing ($\chi^2_{(2)}=1.72$, p=0.424).

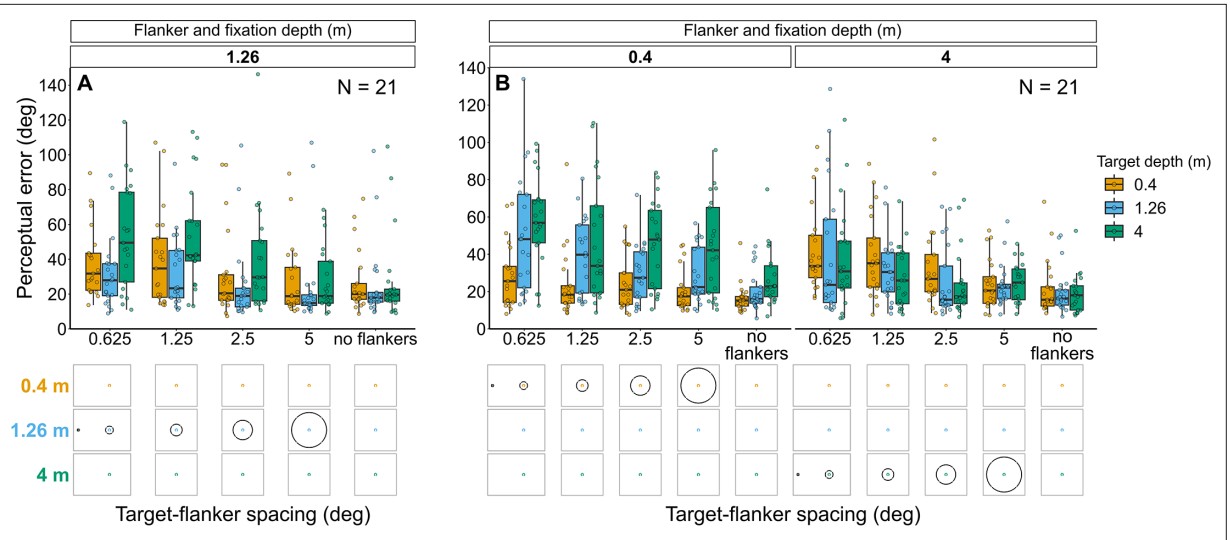

**Figure 6.** Perceptual error results for Experiments 3 (**A**) and 4 (**B**) in which target depth was varied while the flanker ring was always at fixation depth. Box plots show medians plus the interquartile range (IQR), whiskers are the lowest and highest values that are within 1.5 times the IQR from the upper and lower quartiles. Points show the perceptual error for each individual observer. The visual aid below each graph shows the depth of the target (based on the same color key as the box plot) relative to the depth of the fixation point and flanker ring (shown in black) for each stimulus condition. The relative sizes of, and distances between, the fixation point, target, and flanker ring are to scale. Note that the fixation point is only shown for the 0.625° target-flanker spacing due to limited space within the figure.

The online version of this article includes the following figure supplement(s) for figure 6:

**Figure supplement 1.** Perceptual error for Experiments 3 (**A**) and 4 (**B**) calculated only from trials in which the observer reported seeing the target inside the flanker ring.

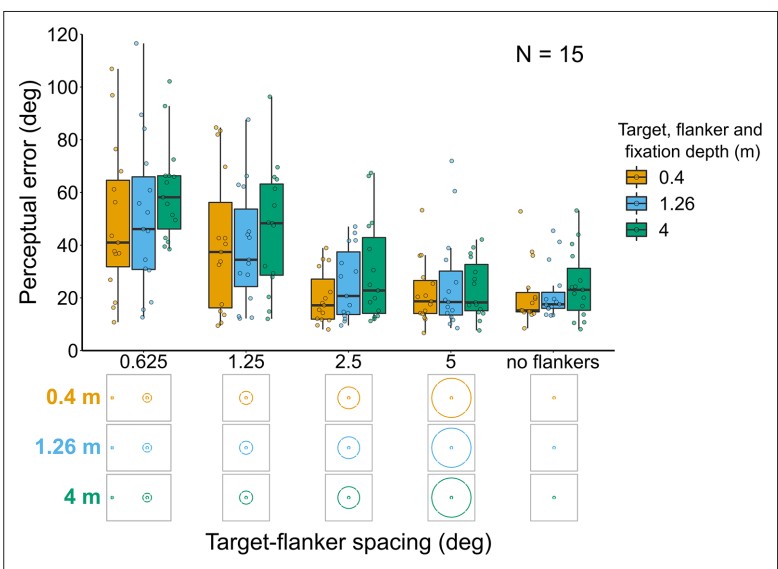

**Figure 7.** Perceptual error results for Experiment 5 in which the target and flanker ring were always presented at the same depth as the fixation point. Box plots show medians plus the interquartile range (IQR), whiskers are the lowest and highest values that are within 1.5 times the IQR from the upper and lower quartiles. Points show the perceptual error for each individual observer. The visual aid below the graph shows the depth of the target, flanker ring, and fixation point (based on the same color key as the box plot) for each stimulus condition. The relative sizes of, and distances between, the fixation point, target, and flanker ring are to scale. Note that the fixation point is only shown for the 0.625° target-flanker spacing so not to overcrowd the figure.

## Perceived target position relative to the flanker ring

### Target always at fixation depth with variable flanker depth

#### Experiment 1: Target and fixation at 1.26 m with flanker ring presented at each depth (Figure 8A and Figure 8—figure supplement 1)

In Experiment 1, most observers consistently reported seeing the target inside the flanker ring. However, when the flanker ring was away from target-fixation depth there was a reduction in the proportion of observers who consistently reported seeing the target within the flanker ring when the target-flanker spacing was small (mixed effects binary logistic regression models compared via LRT; interaction between target-flanker spacing and flanker depth: $\chi^2_{(2)}$=36.12, p<0.001). This effect was greatest when the flanker ring was displayed behind the target and fixation.

Note that in all four experiments, the response to the control (i.e. no flanker condition) indicates the probability of an observer making an inaccurate report (e.g. they selected the wrong option by mistake or missed the stimulus and reported a guess rather than selecting the 'Unsure' option).

#### Experiment 2: Target and fixation at 0.4 m or 4 m with flanker ring presented at each depth (Figure 8B and Figure 8—figure supplement 2)

As with the previous experiment, when the flanker ring was away from target-fixation depth, there was a reduction in the overall proportion of trials in which observers reported seeing the target within the flanker ring (interaction between target-fixation depth and flanker depth: $\chi^2_{(2)}$=374.26, p<0.001). The greatest reduction occurred when target-flanker spacing was smaller (effect of target-flanker spacing: $\chi^2_{(1)}$=178.65, p<0.001). There was no significant interaction between flanker depth and target-flanker spacing ($\chi^2_{(2)}$=0.24, p=0.886) nor between target-fixation depth and target-flanker spacing ($\chi^2_{(1)}$=1.64, p=0.2).

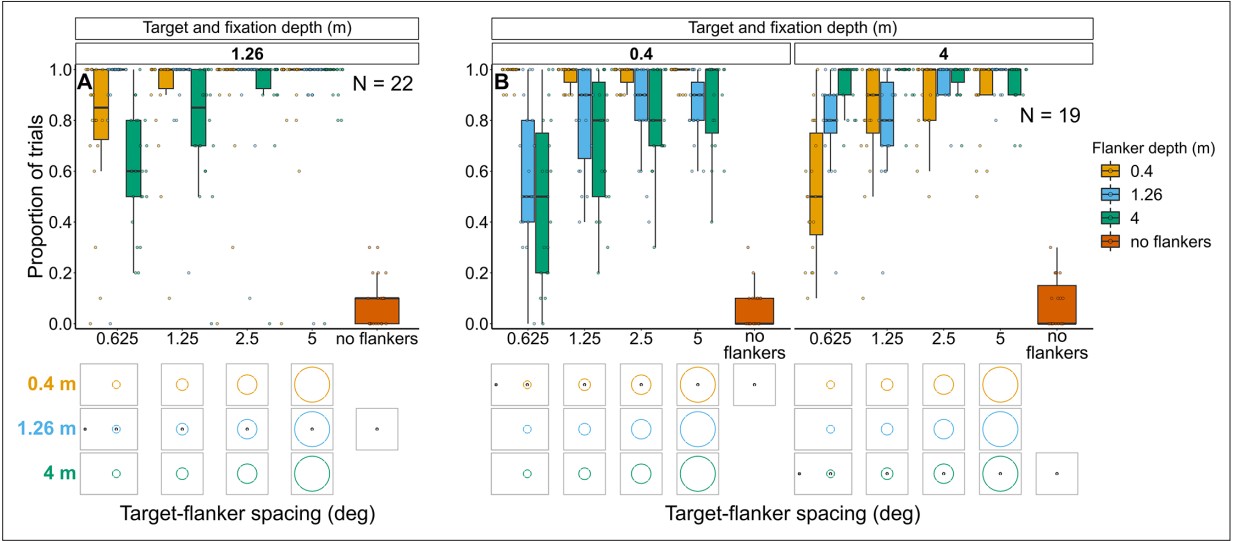

**Figure 8.** Proportion of trials in which observers reported seeing the target inside the flanker ring for Experiments 1 (**A**) and 2 (**B**). In both experiments flanker depth was varied while the target was always at fixation depth. Box plots show medians plus the interquartile range (IQR), whiskers are the lowest and highest values that are within 1.5 times the IQR from the upper and lower quartiles. Points show the proportion of trials for each individual observer. The visual aid below each graph shows the depth of the flanker ring (based on the same color key as the box plot) relative to the depth of the fixation point and target (shown in black) for each stimulus condition. The relative sizes of, and distances between, the fixation point, target, and flanker ring are to scale. Note that the fixation point is only shown for the 0.625° target-flanker spacing due to limited space within the figure.

The online version of this article includes the following figure supplement(s) for figure 8:

**Figure supplement 1.** Perceived target position relative to the flanker ring in Experiment 1.

**Figure supplement 2.** Perceived target position relative to the flanker ring in Experiment 2.

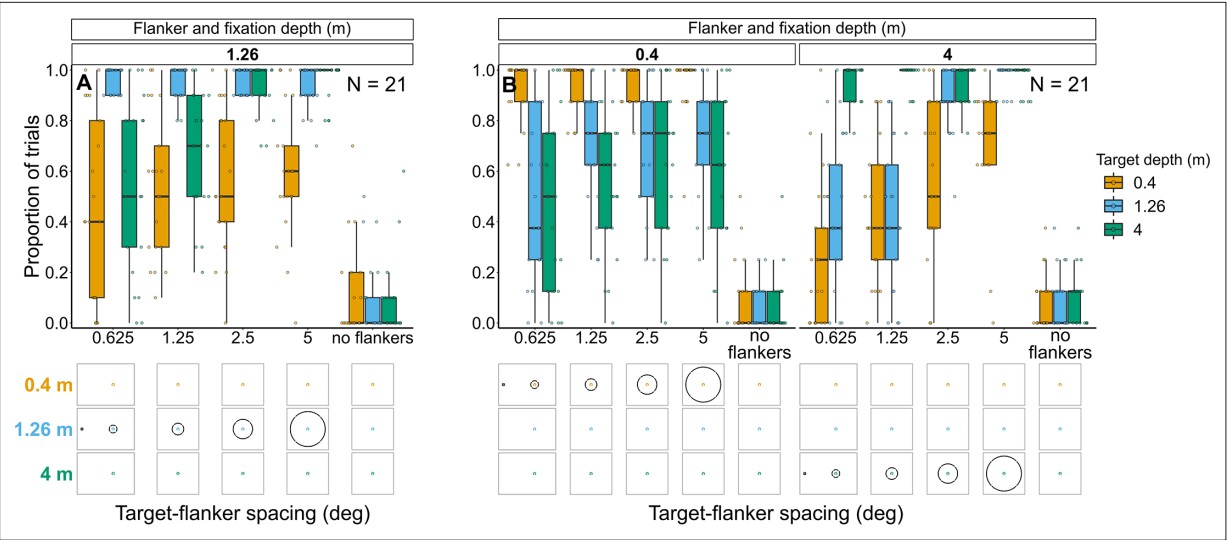

**Figure 9.** Proportion of trials in which observers reported seeing the target inside the flanker ring for Experiments 3 (**A**) and 4 (**B**). In both experiments, target depth was varied while the flanker ring was always at fixation depth. Box plots show medians plus the interquartile range (IQR), whiskers are the lowest and highest values that are within 1.5 times the IQR from the upper and lower quartiles. Points show the proportion of trials for each individual observer. The visual aid below each graph shows the depth of the target (based on the same color key as the box plot) relative to the depth of the fixation point and flanker ring (shown in black) for each stimulus condition. The relative sizes of, and distances between, the fixation point, target, and flanker ring are to scale. Note that the fixation point is only shown for the 0.625° target-flanker spacing due to limited space within the figure.

The online version of this article includes the following figure supplement(s) for figure 9:

**Figure supplement 1.** Perceived target position relative to the flanker ring in Experiment 3.

**Figure supplement 2.** Perceived target position relative to the flanker ring in Experiment 4.

## Flanker ring always at fixation depth with variable target depth

### Experiment 3: Flanker ring and fixation at 1.26 m with target presented at each depth (Figure 9A and Figure 9—figure supplement 1)

When the target was away from flanker-fixation depth, observers were less likely to report seeing the target within the flanker ring. This effect was greatest when target-flanker spacing was smaller and when the target was displayed in front of the flanker ring and fixation (interaction between target-flanker spacing and target depth: $\chi^2_{(2)}$=100.38, p<0.001).

### Experiment 4: Flanker ring and fixation at 0.4 m or 4 m with target presented at each depth (Figure 9 and Figure 9—figure supplement 2)

As with the previous experiment, the greater the difference between target depth and flanker-fixation depth, the less likely observers were to report seeing the target within the flanker ring (interaction between flanker-fixation depth and target depth: $\chi^2_{(2)}$=779.49, p<0.001), particularly when target-flanker spacing was small (interaction between target-flanker spacing and target depth: $\chi^2_{(2)}$=23.27, p<0.001). However, target-flanker spacing had a weaker effect on the proportion of trials in which observers reported seeing the target within the flanker ring when the flanker ring and fixation were near compared to far (interaction between target-flanker spacing and flanker-fixation depth: $\chi^2_{(1)}$=56.23, p<0.001).

## Individual differences in perceptual error and perceived target position in experienced observers

While we were able to detect overall statistically significant trends in the results, it is apparent from *Figures 5–9* that there is a large amount of variation within the data. This raises questions about whether the variation is because the effects of depth are highly variable between some observers, or because the naive observers were not very stable at indicating their perception. The latter could be due to the naive observers being relatively inexperienced at the task, despite receiving training,

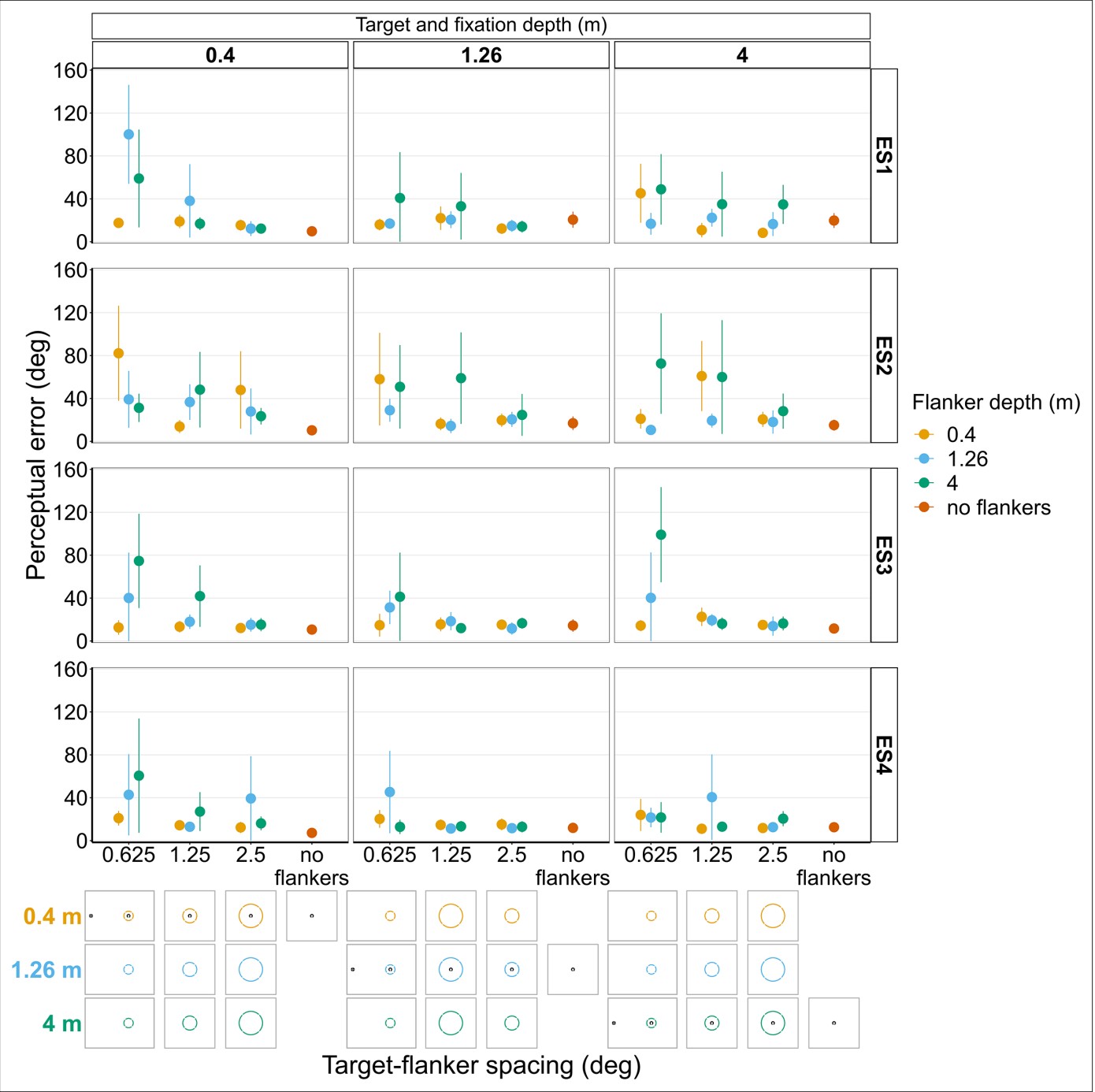

**Figure 10.** Perceptual error results from individual experienced subjects (ES1-4) for Experiments 1 and 2 in which flanker depth was varied while the target was always at fixation depth. Error bars show 95% bootstrapped confidence intervals. Note that the 5° target-flanker spacing condition was not included in this set of experiments. Since the same subjects participated in both experiments, the results from Experiments 1 and 2 have been combined into a single figure to make it easier to compare the different conditions. The visual aid below the graph shows the depth of the flanker ring (based on the same color key as the plot) relative to the depth of the fixation point and target (shown in black) for each stimulus condition. The relative sizes of, and distances between, the fixation point, target, and flanker ring are to scale. Note that the fixation point is only shown for the 0.625° target-flanker spacing due to limited space within the figure.

The online version of this article includes the following figure supplement(s) for figure 10:

**Figure supplement 1.** Proportion of trials in which each individual experienced subject (ES1-4) reported seeing the target inside the flanker ring for Experiments 1 and 2.

**Figure supplement 2.** Perceived target position relative to the flanker ring reported by each individual experienced subject (ES1-4) in Experiments 1 and 2.

since each observer only participated in a single experiment. Therefore, to investigate the effects of depth at the individual level we repeated the study with a small number of highly trained, experienced observers who performed all five experiments (see Methods for details).

## Target always at fixation depth with variable flanker depth

### Experiments 1 and 2 with experienced observers (Figure 10 and Figure 10—figure suppliments 1 and 2)

The small sample size means direct comparisons between these results and the results reported above for the main study should be avoided. Nevertheless, the overall effects reported above are also apparent in most of the experienced observers (*Figure 10*). However, *Figure 10* shows that even among highly trained, experienced observers there is a large amount of variation both between and within subjects. For instance, when the target and fixation are on the near screen (0.4 m) and target-flanker spacing is 0.625°, ES3 and ES4 both exhibit a similar increase in perceptual error with increasing flanker depth, which is consistent with the overall effect shown in *Figure 5B*. In contrast, ES2 shows the opposite trend for the same stimulus condition, a reduction in perceptual error with increased flanker depth. However, when the target and fixation were on the mid screen (1.26 m) and target-flanker spacing is 0.625°, ES2 does exhibit an increase in perceptual error when the flanker ring is behind the target and fixation (as do ES1 and ES3), thus providing an example of between and within subject variation. The variation in perceptual error between subjects cannot solely be explained by differences in the perceived position of the target relative to the flanker ring (*Figure 10—figure supplements 1 and 2*). For example, when the target and fixation were on the near screen and target-flanker spacing was 0.625°, the perceived target positions reported by ES2 and ES4 were remarkably similar (*Figure 10—figure supplement 2*) despite exhibiting the previously mentioned opposite effects of depth on perceptual error (*Figure 10*).

## Flanker ring always at fixation depth with variable target depth

### Experiments 3 and 4 with experienced observers (Figure 11 and Figure 11—figure suppliments 1 and 2)

Like the results from the repeat of Experiments 1 and 2 with the experienced subjects, *Figure 11* shows there was also a relatively large amount of between and within subject variation in Experiments 3 and 4 when repeated with the experienced subjects. For instance, if we look at the analogous stimulus condition to the one used in the previous example, that is, flanker ring and fixation on the near screen (0.4 m) with a target-flanker spacing of 0.625°, ES1, ES3, and ES4 all exhibit an increase in perceptual error with increasing target depth, which is consistent with the overall effect shown in *Figure 6B*. In contrast, ES2 again shows the opposite effect for the same stimulus condition, a reduction in perceptual error with increasing target depth. Again, the variation in perceptual error between subjects cannot solely be explained by differences in the perceived position of the target relative to the flanker ring (*Figure 11—figure supplements 1 and 2*). For instance, the perceived target positions reported by ES2 and ES3 were very similar (*Figure 11—figure supplement 2*) when the flanker ring and fixation were on the near screen and target-flanker spacing was 0.625°, yet the two observers exhibited the opposite effect of increasing target depth on perceptual error (*Figure 11*).

## Target and flanker ring at fixation depth

### Experiment 5 with experienced observers (Figure 12)

*Figure 12* shows there was also some variation between the experienced subjects in Experiment 5 with ES2 and ES3 both exhibiting the highest perceptual error when the target, flanker ring and fixation were furthest away (though this is less pronounced at greater eccentricities for ES3). ES1 and ES4 on the other hand both show a less consistent effect of depth.

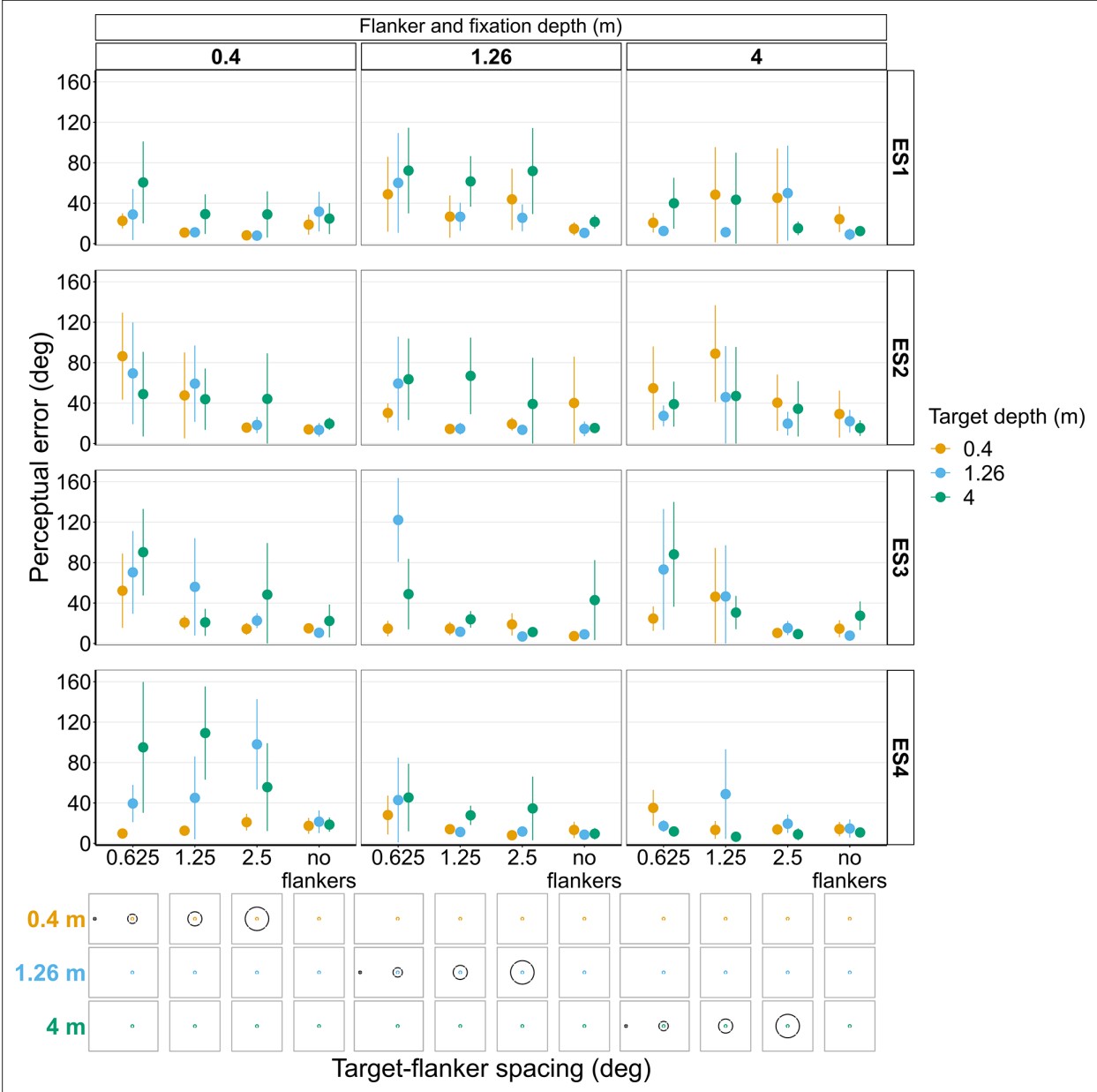

**Figure 11.** Perceptual error results from individual experienced subjects (ES1-4) for Experiments 3 and 4 in which target depth was varied while the flanker ring was always at fixation depth. Error bars show 95% bootstrapped confidence intervals. Note that the 5° target-flanker spacing condition was not included in this set of experiments. Since the same subjects participated in both experiments, the results from Experiments 3 and 4 have been combined into a single figure to make it easier to compare the different conditions. The visual aid below each graph shows the depth of the target (based on the same color key as the plot) relative to the depth of the fixation point and flanker ring (shown in black) for each stimulus condition. The relative sizes of, and distances between, the fixation point, target, and flanker ring are to scale. Note that the fixation point is only shown for the 0.625° target-flanker spacing due to limited space within the figure.

The online version of this article includes the following figure supplement(s) for figure 11:

**Figure supplement 1.** Proportion of trials in which each individual experienced subject (ES1-4) reported seeing the target inside the flanker ring for Experiments 3 and 4.

**Figure supplement 2.** Perceived target position relative to the flanker ring reported by each individual experienced subject (ES1-4) in Experiments 3 and 4.

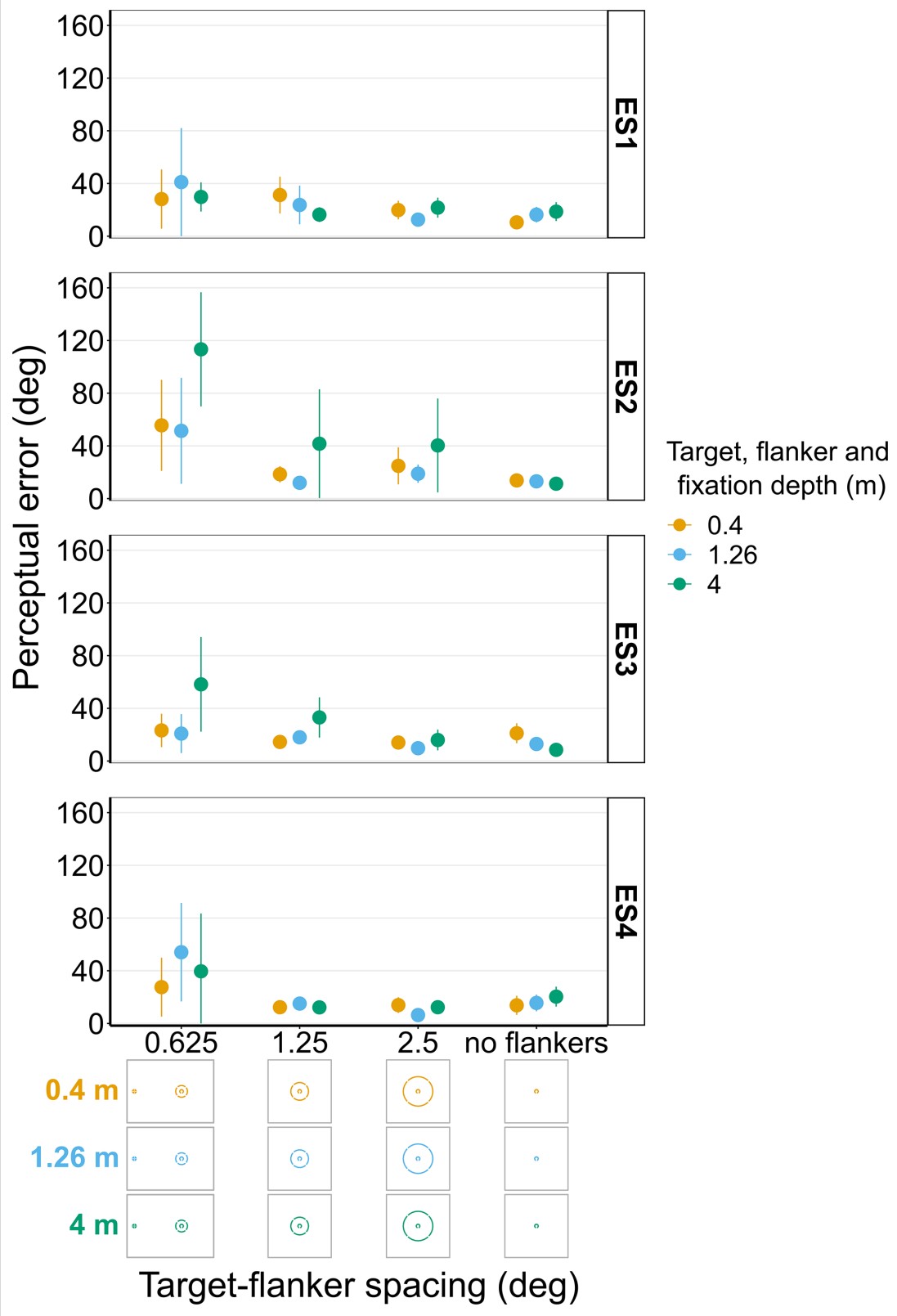

**Figure 12.** Perceptual error results from individual experienced subjects (ES1-4) for Experiment 5 in which the target and flanker ring were always presented at the same depth as the fixation point. Error bars show 95% bootstrapped confidence intervals. Note that the 5° target-flanker spacing condition was not included in this set of experiments. The visual aid below the graph shows the depth of the target, flanker ring, and fixation point (based on the same color key as the plot) for each stimulus condition. The relative sizes of, and distances between, the fixation point, target, and flanker ring are to scale. Note that the fixation point is only shown for the 0.625° target-flanker spacing so not to overcrowd the figure.

## Discussion

In all experiments, perceptual error generally decreased as the target-flanker spacing increased, which is in line with previous crowding research (*Bouma, 1970*; *Pelli, 2008*). Overall, perceptual error was higher when the target or flanker ring were presented at a different depth from the other and from fixation. The greatest differences in perceptual error among the different depth conditions occurred when the target or flanker ring were presented behind fixation depth. In contrast, when the target or flanker ring were presented in front of fixation depth, the differences in perceptual error among depth conditions were notably smaller. The results also show that when the target, flanker ring and fixation were all at the same depth plane, crowding tended to increase with the absolute distance of the stimuli from the observer. However, in all the main experiments with naive observers there was a large amount of variation within the data. Follow up experiments with a small sample of highly trained, experienced observers were largely consistent with the main effects observed in the main study. Moreover, the experienced observers also exhibited a high degree of between subject variation, which supports the suggestion that the variability in the main data from the large sample of naive observers is representative of the natural variation between different individuals. Therefore, although significant effects were found at the population level, the effects of large differences in target-flanker depth can be highly variable at the individual level. This demonstrates the importance of using a large sample size to investigate effects that are not necessarily detectable at the individual level. The discussion of the findings will focus on the overall results from the main study on naive participants, with the findings from the small sample of experienced observers used to provide additional context and insight.

The main finding that a depth difference between the flankers and the target led to an increase, rather than a decrease, in perceptual error is different from the findings of several previous studies that investigated the effect of depth on crowding using a stereoscopic presentation (*Astle et al., 2014*; *Felisberti et al., 2005*; *Kooi et al., 1994*; *Sayim et al., 2008*). For example, *Astle et al., 2014* found that when the target was presented at fixation depth (comparable to Experiments 1 and 2 in our study), increasing the disparity between the target and flankers resulted in less crowding, and ultimately a complete release from crowding. The authors also reported that crowding was greater when flankers were presented in crossed disparity (in front of the target and fixation), compared to uncrossed disparity (behind the target and fixation) (*Astle et al., 2014*). Similarly, when the flankers were presented at fixation depth (comparable to Experiments 3 and 4 in our study), *Sayim et al., 2008* found that foveal crowding was lower when the target was presented in front of or behind the flankers and fixation.

Several explanations for crowding have been proposed in which crowding occurs because target and flanking features are inappropriately grouped (*Banks et al., 1979*; *Herzog and Manassi, 2015*; *Pachai et al., 2016*). Consistent with this explanation, crowding can be alleviated when the flankers are grouped together and segmented from the target (*Herzog and Manassi, 2015*). One piece of evidence in support of this account is the reduction of crowding that can occur when flankers are presented at a different depth from the target (*Astle et al., 2014*; *Felisberti et al., 2005*; *Kooi et al., 1994*; *Sayim et al., 2008*). Our results challenge this view of crowding because it would predict that increasing the depth difference between target and flankers would decrease the probability of them being inappropriately grouped and increase the probability of the flankers being segmented from the target, thereby decreasing perceptual error, the opposite of our finding. Other models of crowding, such as texture processing and feature integration (*Balas et al., 2009*; *Freeman and Simoncelli, 2011*; *Parkes et al., 2001*; *van den Berg et al., 2012*; *Wallis and Bex, 2012*), sampling errors (*Ester et al., 2015*; *Ester et al., 2014*; *Harrison and Bex, 2017*; *Harrison and Bex, 2016*; *Harrison and Bex, 2015*), attentional resolution limits (*He et al., 1996*) or saccade properties (*Nandy and Tjan, 2012*) are agnostic concerning the influence of depth differences among targets and flankers. Therefore, each of these models would require modification to explain the increase in crowding that we observe with large real depth differences between a target and its flankers.

In contrast with studies that controlled depth with small stereoscopic disparities, our results do agree with the findings of *Eberhardt and Huckauf, 2020* who also used a real-depth display and tested larger differences in depth between the target and flankers than previous studies had examined, though depth differences were still much smaller than those tested in our current study. When the target was at fixation depth (comparable to Experiments 1 and 2 in our study), *Eberhardt and*

*Huckauf, 2020* found that crowding increased with increasing target-flanker depth difference. The authors also found that crowding was greater when the flankers were behind the target and fixation compared to in front of it (*Eberhardt and Huckauf, 2020*). Our findings from Experiments 1 and 2 are consistent with these results. However, when the flankers were at fixation depth (comparable to Experiments 3 and 4 in our study), *Eberhardt and Huckauf, 2020* found that crowding was higher when the target was in front of the flankers and fixation compared to when it was behind them. In contrast, in Experiments 3 and 4 we found that perceptual error was greatest when the target was presented behind the flanker ring and fixation rather than in front of it. This difference is likely due to a combination of high defocus blur when the target was behind fixation combined with the finding from Experiment 5 which found that in our study a larger absolute distance from the observer resulted in higher perceptual error.

Why the inconsistencies across studies? It is important to consider that most of the previous studies used very small depth differences, the largest being 0.1 diopters in *Eberhardt and Huckauf, 2019*; *Eberhardt and Huckauf, 2020*. In comparison, the depth difference in our study ranged from 0.54 diopters (between the mid and far screens) to 2.25 diopters (between the near and far screens). Our results, together with those from previous studies, suggest that small differences in depth between the target and flankers reduce crowding (*Astle et al., 2014*; *Felisberti et al., 2005*; *Kooi et al., 1994*; *Sayim et al., 2008*), while larger differences in depth can increase it (*Eberhardt and Huckauf, 2020* and results from our study). Although this pattern is quite surprising, there is precedent for non-linear effects of depth on perceptual processing outside the crowding literature. In a multiple object tracking task, when the targets and distractors were distributed equally on two depth planes, tracking accuracy was better than the accuracy during trials in which targets and distractors appeared on a single depth plane when the depth difference between planes was small (~6 cm) (*Ur Rehman et al., 2015*; *Viswanathan and Mingolla, 2002*), but worse when the depth difference between planes was large (50 cm) (*Ur Rehman et al., 2015*).

How could depth affect perception in this nonlinear way? The geometry of the eyes and the 3D environment create interocular disparity differences between the images in each eye. When corresponding features in the images in each eye can be identified, the images can be fused to generate a binocular perceptual experience of depth (for review see *Nityananda and Read, 2017*). There is a horizontal and vertical spatial limit to the interocular disparity difference over which the visual system attempts to identify corresponding retinal points in order to fuse different 2D interocular images as a single 3D object (*Westheimer, 1979*). This 2D spatial correspondence limit creates a 3D volume along the horopter known as Panum's Fusional Area (*Panum, 1858*). Outside Panum's Fusional Area, there is a loss of stereoscopic depth perception and either double vision (diplopia) may be experienced, or single vision may occur if the image from one eye is partly suppressed (*Georgeson and Wallis, 2014*; *Ono et al., 1977*; *Spiegel et al., 2016*; *Stidwill and Fletcher, 2010*). The spatial extent of Panum's Fusional Area depends on the spatial frequency of the stimuli (*Maiello et al., 2020*; *Pulliam, 1981*; *Reynaud and Hess, 2017*; *Wilcox and Allison, 2009*; *Yang and Blake, 1991*) and increases with eccentricity (*Hampton and Kertesz, 1983*; *Harrold and Grove, 2019*; *Qin et al., 2006*).

It is possible that differences between fusion and depth perception within Panum's Fusional Area versus diplopia, and/or local suppression for single vision, outside of it can explain the patterns of results observed between studies. Small disparity differences between target and flankers create differences in perceived depth that can alleviate the effects of crowding, potentially through ungrouping of target and flanking elements (*Herzog et al., 2015*; *Herzog and Manassi, 2015*; *Manassi et al., 2012*) or by independent processing of target and flankers in different depth-selective channels (*Norcia et al., 1985*; *Pulliam, 1981*; *Reynaud and Hess, 2017*; *Tyler et al., 1994*; *Wilcox and Allison, 2009*; *Yang and Blake, 1991*) by any of the computational models of crowding based on texture processing and feature integration (*Balas et al., 2009*; *Freeman and Simoncelli, 2011*; *Parkes et al., 2001*; *van den Berg et al., 2012*; *Wallis and Bex, 2012*), sampling errors (*Ester et al., 2015*; *Ester et al., 2014*; *Harrison and Bex, 2017*; *Harrison and Bex, 2016*; *Harrison and Bex, 2015*) attentional resolution limits (*He et al., 1996*) or saccade properties (*Nandy and Tjan, 2012*). Large interocular disparity differences on the other hand can induce diplopia that could increase the perceived number of flankers (*Eberhardt and Huckauf, 2019*). In theory, this perceived increase in the number of flanking elements could result in higher perceptual error because increasing the number of flankers is known to increase crowding under certain conditions (*Chanceaux et al., 2014*; *Manassi*

*et al., 2012*; *Pluháček et al., 2021*). It is however worth noting that the opposite can occur, that is reduced crowding with increasing number of flankers, in cases where the additional flankers are grouped together (*Manassi et al., 2013*; *Manassi et al., 2012*). Surprisingly, when asked, only a few of the naive observers reported experiencing diplopia, and those that did reported only experiencing it in a minority of presentations for which they selected the 'unsure or no ring' option (see Methods) when reporting the perceived location of the target (*Figure 8—figure supplements 1E and 2E*, and *Figure 9—figure supplements 1E and 2E*). However, among the four experienced observers, three verbally reported experiencing diplopia during some of the conditions for which they either selected the 'unsure or no ring' option (*Figure 10—figure supplement 2E* and *Figure 11—figure supplement 2E*), or in cases when the diplopia was barely visible, they chose the option that was the closest match to the clearest stimulus image. The fact that the experienced observers were more likely to report experiencing diplopia, possibly because they were better at recognizing it, raises the possibility that diplopia may have been more common among the naive observers than was indicated by asking them. However, it is worth noting that one of the four experienced observers reported that they never experienced diplopia in any of the experiments, which is more consistent with the majority of naive observers. It is possible that the short presentation time of the stimulus was too fast for most observers to notice diplopia consciously (i.e. diplopia did not enter awareness), or that diplopia was suppressed entirely by local suppression to maintain single vision. Attention effects such as inattentional blindness (*Bredemeier and Simons, 2012*; *Simons and Chabris, 1999*) could also play a role in explaining why so few observers reported diplopia in the main experiments. There is conflicting evidence as to whether conscious awareness of a flanker is necessary for it to induce crowding (compare *Ho and Cheung, 2011* vs *Wallis and Bex, 2011*), and thus whether 'unconscious diplopia' could have increased crowding in this study due to the reason highlighted above is a matter for debate and beyond the scope of our findings. In addition, it is worth noting that due to the lack of binocular fusion of stimulus areas outside Panum's Fusional Area, we can't rule out the possible involvement of binocular rivalry which has been shown to interfere with object recognition not just in isolation but also when combined with crowding (*Kim et al., 2013*).

Another explanation for higher perceptual error with increasing target-flanker depth differences could be perceived target misalignment. As explained in the Methods, at the start of Experiments 1–4 each observer performed a subjective spatial calibration to ensure that the angular positions of the stimuli and fixation point relative to primary gaze were consistent across all three depth planes. Although the main experiments were performed under binocular viewing conditions, this spatial calibration had to be performed monocularly using the observer's dominant eye to avoid confusion caused by diplopia or rivalry (see Methods). This means that the screens were imperfectly aligned for the non-dominant eye. Therefore, as in the real world, at certain flanker sizes and separations there may be occlusions between target and flanker features that could affect target report accuracy, which is not the same mechanism as crowding. In our study, such misalignment could result from local suppression of the calibrated eye to maintain single vision, and/or head movements following the spatial calibration. To examine how target misalignment affected the overall results, we recalculated perceptual error for the main experiments with naive observers using only trials in which the observer reported seeing the target inside the flanker ring. We then reanalyzed these data the same way as for perceptual error calculated from the full data set (see Methods). Apart from Experiment 2, the effect of flanker and/or target depth remained statistically significant (see *Figure 5—figure supplement 1* and *Figure 6—figure supplement 1* for plots and analysis results). Therefore, across the majority of experiments, misalignment or occlusion of the target by the flanker ring is not enough to explain the overall results. This supports the suggestion that the increase in perceptual error when the target and flankers are at different depths is being driven by an increase in crowding.

This current study is the first to investigate how crowding is affected by large, real differences in depth between a target and flanking features. This is important because objects in the real world are often flanked by visual features that vary greatly in their distance from an observer, and many will fall outside the limits of Panum's Fusional Area (*Figure 1*). A limitation of our approach is that the use of relatively simple stimuli in a tightly controlled experiment limits the generalizability of our findings to the real world where clutter is abundant and more complex. Future research should therefore aim to utilize stimuli that better capture the complexity and heterogeneity of natural scenes. Future work should also investigate if the reported effects of depth on crowding in our study hold at different

target eccentricities, particularly given that the range of Panum's Fusional Area changes with eccentricity (*Hampton and Kertesz, 1983*; *Harrold and Grove, 2019*; *Qin et al., 2006*). Nevertheless, our findings still show that crowding from clutter outside Panum's Fusional Area has the potential to significantly impact object recognition and visual perception in the peripheral field. Our findings therefore do not support the view that depth differences between targets and flankers necessarily reduce crowding, instead the effect of depth on crowding appears to be more complex than originally thought. Furthermore, data collected from experienced participants shows that the overall trends at the population level may not always be apparent at the individual level. Nonetheless, our results, alongside those of previous studies, suggest that small differences in depth between targets and flankers do not have the same effect on crowding as larger differences in depth in which either the target or flankers are presented outside the limits of binocular fusion. This finding confirms the potential importance and relevance of crowding in three-dimensional scenes.

## Methods

### Subject recruitment

For the main study, a total of 110 naive subjects were recruited from the Northeastern University undergraduate population in exchange for course credit as compensation for their time. Of these, seven were excluded immediately following screening because they did not meet the inclusion criteria of normal or corrected-to-normal vision. An additional five subjects were excluded because of issues with the experimental code (N=3), calibration issues (N=1) or because they did not follow the task instructions (N=1). Thus, a total of 98 subjects (32 males, 66 females; mean age = 18.86 years) contributed usable data across five separate experiments (22 in Experiment 1, 19 in Experiment 2, 21 in Experiments 3 and 4, and 15 in Experiment 5).

In addition to the naive subjects who each participated in one of the five experiments as part of the main study, we also separately tested four 'experienced subjects' (2 males, 2 females; mean age = 29 years) who participated in all five experiments in a random order. All the experienced subjects (none of whom were the authors) were volunteers from within the university vision research community and were therefore extremely familiar and well-practiced in participating in psychophysics experiments such as those in our current study. The purpose of repeating the study with a small group of highly trained, experienced subjects who participated in all five experiments was to enable us to better explore the effects of depth at the individual level.

All subjects read and signed an informed consent form approved by the Northeastern University Ethics Board before the experiment began. The experimental procedure was approved by the institutional review board at Northeastern University, and the experiments were performed in accordance with the tenets of the Declaration of Helsinki.

### Multi-depth plane display setup

The study utilized a multi-depth plane display consisting of three screens at 0.4 m (near screen), 1.26 m (mid screen), and 4 m (far screen) from the observer (*Figure 2*). For the far screen, we used an ultra-short throw projector (Bomaker Polaris 4 K Laser Projector) projected onto a 240 cm x 250 cm white wall at the end of the room at a throw distance (measured horizontally from the projector to the wall) of approximately 50 cm. The mid and near screens were made from 101.5 cm x 76 cm and 46 cm x 36.5 cm clear 5 mm acrylic, respectively, held in position by a frame constructed from extruded aluminum rails. The transparent acrylic was covered on one side with ClearBright transparent display film (Lux Labs Inc, Boston, MA, USA). The nanoparticle film enabled us to display images from two projectors (InFocus IN3116) onto the near and mid screens while preserving their transparency (*Hsu et al., 2014*). This design enables observers to simultaneously view adjacent stimuli presented on all three depth planes at the same time. The throw distances of the near and mid screen projectors (measured from the projector to the bottom of the screen) were approximately 72 cm and approximately 150 cm, respectively. Both projectors were positioned and angled so that projected light that passed through the transparent screens did not fall upon the observer or any areas visible by the observer, such as an adjacent screen. Key stoning was used to digitally correct distortions in the image caused by the projection angle of the near and mid screen projectors. The screens were temporally synchronized and spatially aligned to ensure display timing and position were consistent

across the three depth planes. All experiments were conducted in Python using PsychoPy (v2021.2.3 for experiments with the naive observers and v2022.2.5 for the experiments with the four experienced observers) (*Peirce et al., 2019*).

## Spatial calibration

Each observer performed a subjective spatial calibration at the start of Experiments 1–4 to ensure that the angular positions of the stimuli and fixation point relative to primary gaze were the same (or as close as realistically possible) across all three depth planes. The center of the far screen (which was at primary gaze when the observers head was positioned in the chin and head rest) was used as the fixed reference against which all three screens were calibrated. With their head in the chin and head rest, observers were instructed to sequentially position a series of targets so that they were in the same position as a fixed reference target that was displayed on the reference screen. The calibration was performed under monocular conditions using the observer's dominant eye as determined by the Miles test (*Miles, 1929*). The calibration was performed monocularly because during a preliminary experiment subjects reported finding the task too difficult to complete under binocular conditions (e.g., because of diplopia, an issue that is considered further in the Discussion section). For Experiment 5, the spatial calibration was performed by the experimenter prior to the experiment using a webcam positioned at the observer's cyclopean point. The main experiment was conducted under binocular viewing conditions and observers were instructed to maintain the same head position within the chin and head rest throughout the experiment.

**Table 1.** Summary of the stimulus conditions in each experiment. Note that for target-flanker spacing, 'infinity' = no flanker ring. For the experiments conducted with the four experienced subjects, the 5° target-flanker eccentricity condition was not included.

| Experiment | Fixation depth (m) | Target depth (m) | Flanker depth (m) | Target-flanker spacing (degrees) | No. of stimulus conditions | Total no. presentations | Sample size |
|---|---|---|---|---|---|---|---|
| | Mid (1.26) | Mid (1.26) | Near (0.4) | 0.625°, 1.25°, 2.5°, 5° | | | |
| | Mid (1.26) | Mid (1.26) | Mid (1.26) | 0.625°, 1.25°, 2.5°, 5°, infinity | | 130 | |
| Exp 1 | Mid (1.26) | Mid (1.26) | Far (4) | 0.625°, 1.25°, 2.5°, 5° | 13 | (10 reps per stimulus) | N=22 |
| | Near (0.4) | Near (0.4) | Near (0.4) | 0.625°, 1.25°, 2.5°, 5°, infinity | | | |
| | Near (0.4) | Near (0.4) | Mid (1.26) | 0.625°, 1.25°, 2.5°, 5° | | | |
| | Near (0.4) | Near (0.4) | Far (4) | 0.625°, 1.25°, 2.5°, 5° | | | |
| | Far (4) | Far (4) | Near (0.4) | 0.625°, 1.25°, 2.5°, 5° | | | |
| | Far (4) | Far (4) | Mid (1.26) | 0.625°, 1.25°, 2.5°, 5° | | 260 | |
| Exp 2 | Far (4) | Far (4) | Far (4) | 0.625°, 1.25°, 2.5°, 5°, infinity | 26 | (10 reps per stimulus) | N=19 |
| | Mid (1.26) | Near (0.4) | Mid (1.26) | 0.625°, 1.25°, 2.5°, 5°, infinity | | | |
| | Mid (1.26) | Mid (1.26) | Mid (1.26) | 0.625°, 1.25°, 2.5°, 5°, infinity | | 150 | |
| Exp 3 | Mid (1.26) | Far (4) | Mid (1.26) | 0.625°, 1.25°, 2.5°, 5°, infinity | 15 | (10 reps per stimulus) | N=21 |
| | Near (0.4) | Near (0.4) | Near (0.4) | 0.625°, 1.25°, 2.5°, 5°, infinity | | | |
| | Near (0.4) | Mid (1.26) | Near (0.4) | 0.625°, 1.25°, 2.5°, 5°, infinity | | | |
| | Near (0.4) | Far (4) | Near (0.4) | 0.625°, 1.25°, 2.5°, 5°, infinity | | | |
| | Far (4) | Near (0.4) | Far (4) | 0.625°, 1.25°, 2.5°, 5°, infinity | | | |
| | Far (4) | Mid (1.26) | Far (4) | 0.625°, 1.25°, 2.5°, 5°, infinity | | 240 | |
| Exp 4 | Far (4) | Far (4) | Far (4) | 0.625°, 1.25°, 2.5°, 5°, infinity | 30 | (8 reps per stimulus) | N=21 |
| | Near (0.4) | Near (0.4) | Near (0.4) | 0.625°, 1.25°, 2.5°, 5°, infinity | | | |
| | Mid (1.26) | Mid (1.26) | Mid (1.26) | 0.625°, 1.25°, 2.5°, 5°, infinity | | 150 | |
| Exp 5 | Far (4) | Far (4) | Far (4) | 0.625°, 1.25°, 2.5°, 5°, infinity | 15 | (10 reps per stimulus) | N=15 |

## Stimuli and experimental procedure

The target and flankers were inspired by those used in previous studies (*Harrison and Bex, 2017*; *Harrison and Bex, 2015*). The target was a white Landolt-C displayed against a black background at 10° eccentricity to the right or left of the fixation point at primary gaze. Following the structure of a Landolt-C optotype, the target was 1° in diameter with a 0.25° line width and a 0.25° opening. The opening orientation of the target was random each trial, drawn from a uniform 360° distribution. The target was surrounded by a white flanker ring formed of four circular arcs (*Figure 3*). In each depth combination described below, the outer edge of the target was separated from the inner edge of the flanker ring by 5°, 2.5°, 1.25° or 0.625°. These distances of separation represented 1/2, 1/4, 1/8, and 1/16 of the target eccentricity, respectively. Note that for the experiments conducted with the four experienced subjects, the 5° target-flanker eccentricity condition was not included. The line width and size of the four gaps in the flanker ring were kept constant (0.25°). For each target depth, a control condition with no flanker ring was also included. The fixation point consisted of a 1° white circle with a black cross (*Figure 3*). This design was used because it has been reported to elicit more stable fixation compared to a plain circle or cross hair alone (*Thaler et al., 2013*). The background on all screens was always set to the lowest RGB (i.e. $R=G=B=0$) value which was transparent on the near and mid screens, and black on the far screen. Since the mid screen was dimmest ($R=G=B=255$, white = 13.56 cd/m$^2$), the luminance of stimuli displayed on the near and far screens were calibrated to match the luminance of the mid screen using a Photo Research PR-650 spectroradiometer (Photo Research, Chatsworth, CA, USA). The room lights were switched off for the duration of the experiment.

A summary of the stimulus conditions tested in each experiment is provided in *Table 1*. In Experiment 1, the target and fixation point were always on the mid screen while the flanker ring was displayed in front, behind or at fixation depth. In Experiment 2, the target and fixation point were presented on the near or far screen while the depth of the flanker ring was varied. Experiments 3 and 4 were the same as Experiments 1 and 2 respectively except that the flanker ring was presented at fixation depth while the target depth was varied. For Experiment 5, the target, flanker ring and fixation were presented together on the same screen at three different depths. This was to investigate whether there was an effect of absolute viewing distance on crowding. Each condition was repeated 10 times in Experiments 1, 2, 3 and 5 and eight times in Experiment 4, with equal number of repeats on either side of the fixation point. The order of testing was randomized.

An example trial is illustrated in *Figure 4*. Briefly, at the start of each trial, observers were instructed to focus on the central fixation point. The fixation point appeared on the fixation screen 1 s before stimulus onset in Experiments 1, 3 and 5, and 1.5 s before stimulus onset in Experiments 2 and 4. The fixation point was displayed for slightly longer before stimulus onset in Experiments 2 and 4 to ensure that observers had adequate time to refocus on the fixation point when it switched between the near and far screens. The target and flanker ring (if present) were displayed for 0.25 s. The fixation point remained on the screen for 1 s before being replaced by a randomly oriented Landolt-C that was the same size as the target Landolt-C. The observer used the mouse to rotate this Landolt-C until it matched the perceived orientation of the target and then clicked to report the orientation.

In all experiments except Experiment 5, observers were also asked to report the perceived location of the target relative to the position of the flanker ring. There were five options to choose from, 'target in center of ring', 'target inside ring but not in the center', 'ring obstructs target', 'target outside ring', and 'unsure or no ring'. Observers were instructed to select the option that most closely matched how they perceived the stimulus. If an observer missed the stimulus or they were unsure of the targets' location relative to the flanker ring, for example, because of confusion caused by diplopia, they were instructed to select the 'unsure or no ring' option. Collecting data on the perceived target-flanker location enabled us to examine how differences in target-flanker depth influenced the perception of the stimulus and investigate the potential effect that factors such as target misalignment and possible diplopia may have had on perceptual error (see Discussion). Following this selection, the next trial started immediately.

Before starting an experiment, all participants underwent training during which they completed as many practice trials as they needed to familiarize themselves with the task. The main experiment only began once the researcher running the study was confident that the participant understood the task and was able to perform it correctly. These practice trials did not contribute to the collected data.

## Data processing and statistical analysis

### Perceptual error

To quantify crowding for each stimulus condition, we first calculated the report error for each presentation, that is, the angular difference between the reported gap orientation of the target and the actual gap orientation. For each subject, this generated 10 report errors per stimulus condition in Experiments 1, 2, 3, and 5, and eight per stimulus condition in Experiment 4. We then used these report errors to calculate the circular standard deviation for each stimulus condition using the Circular Statistics Toolbox (*Berens, 2009*) in MATLAB R2022a (MathWorks Inc, Natick, MA, USA). We refer to this measure as perceptual error, which is reported in degrees.

For the main study with naive observers, the effects of target/flanker depth and target-flanker spacing on perceptual error were analyzed using mixed effects multiple linear regression, fitted using the lme4 package (*Bates et al., 2015*) in R (*R Development Core Team, 2022*). Perceptual error was used as the response (dependent) variable after being natural log transformed to correct for positive skew. For all five experiments, we included target-flanker spacing as a fixed effect. We also included flanker depth for Experiments 1 and 2, target depth for Experiments 3 and 4 or target-flanker-fixation depth for Experiment 5. Target-fixation depth or flanker-fixation depth were also included as a third fixed effect for Experiments 2 and 4, respectively. All possible pairwise interactions of fixed effects were fitted in the full model. Subject identification (ID) was included as a random effect in all experiments to control for repeated measures. We first used model simplification, whereby models were compared with one another using a likelihood ratio test (LRT), to remove non-significant interactions sequentially. This enabled us to test the effect of individual fixed effects that were not involved in an interaction. We then used the minimum model (i.e. model containing all fixed effects of interest plus any significant two-way interactions) to report the significance of each main effect or significant interaction(s) by using a LRT to compare the minimum model with the same model but with the effect/interaction of interest removed. Individual main effects that were part of a significant interaction were not tested in this way because their effect was dependent on another variable. Note that data for the control condition (i.e. no flanker ring) were excluded from statistical analysis. This was to ensure that (a) the fixed effects remained independent and (b) the effect of having no flanker ring did not mask the effect of flanker and target depth.

### Perceived target position relative to flanker ring

We used the observers' perceived target position relative to the flanker ring to calculate the proportion of trials in which observers reported seeing the target inside the flanker ring (i.e. they chose either 'target in center of ring' or 'target inside ring but not in the center'). For statistical analysis, these data were treated as a binary response variable whereby trials in which observers reported seeing the target inside the flanker ring were assigned a value of 1 while all other trials (i.e. those for which the observer chose 'ring obstructs target', 'target outside ring' or 'unsure or no ring') were assigned a value of 0.

For the main study with naive observers, these data were then analyzed using mixed effects binary logistic regression, fitted using the lme4 package (*Bates et al., 2015*) in R (*R Development Core Team, 2022*). We used the DHARMa package (*Hartig, 2022*) to run residual diagnostics that are often overlooked in logistic regression. Perceived target position was used as the binary response variable (i.e. 1=target inside ring and 0=target not inside ring). The fixed effects and pairwise interactions for each experiment were the same as those described above for the analysis of perceptual error. Subject ID was included as a random effect in all experiments to control for repeated measures. Model simplification (to remove non-significant interactions) and significance testing was the same as described above for the analysis of perceptual error. Data for the control condition (i.e., no flanker ring) were excluded from statistical analysis for the same reasons as those outlined above. The separate results showing the proportion of trials for which observers chose each of the five choices for perceived target position relative to the flanker ring can be found in the figure supplements (see Results).

## Acknowledgements

We thank members of the Bex lab and Eskew lab at Northeastern University for valuable feedback during pilot experiments. We also thank MiYoung Kwon and Jan Skerswetat for helpful discussions and

the reviewers for their insightful feedback. This work was supported by NIH grant R01EY032162 (PJB). The Northeastern University IRB approved procedures (IRB #14-09-16), and all observers consented to participation. The two images in *Figure 1* were generated in Unity (Unity Technologies, San Francisco, CA) using 3D models downloaded from Sketchfab (Sketchfab Inc, Paris, France) and licensed under Creative Commons Attribution (CC BY 4.0 https://creativecommons.org/licenses/by/4.0/). The 3D models were "Construction Sign" by Kyle Burton (https://skfb.ly/6WS8O), "Nathan Animated 003 - Walking 3D Man" by Renderpeople (https://skfb.ly/6SO7F) and "American Road Intersection" by jimbogies (https://skfb.ly/o7z8V). The downloaded 3D models were modified in Blender (Blender Foundation, Amsterdam, Netherlands). Special thanks go to Kerri Walter for her help creating the images for *Figure 1*.

## Additional information

### Funding

| Funder | Grant reference number | Author |
| --- | --- | --- |
| National Institutes of Health | R01EY032162 | Peter J Bex |

The funders had no role in study design, data collection and interpretation, or the decision to submit the work for publication.

### Author contributions

Samuel P Smithers, Conceptualization, Data curation, Software, Formal analysis, Supervision, Validation, Investigation, Visualization, Methodology, Writing – original draft, Project administration, Writing – review and editing; Yulong Shao, James Altham, Investigation; Peter J Bex, Conceptualization, Resources, Formal analysis, Supervision, Funding acquisition, Methodology, Writing – original draft, Project administration, Writing – review and editing

### Author ORCIDs

Samuel P Smithers https://orcid.org/0000-0002-2781-7067
Peter J Bex https://orcid.org/0000-0001-7561-7695

### Ethics

Human subjects: The experimental procedure was approved by the institutional review board at Northeastern University (IRB #14-09-16) and the experiments were performed in accordance with the tenets of the Declaration of Helsinki. All subjects read and signed an informed consent form before taking part in the study.

### Decision letter and Author response

Decision letter https://doi.org/10.7554/eLife.85143.sa1
Author response https://doi.org/10.7554/eLife.85143.sa2

## Additional files

### Supplementary files
• MDAR checklist

### Data availability
All data and code required to replicate the analyses in this paper is available on Zenodo.

The following dataset was generated:

| Author(s) | Year | Dataset title | Dataset URL | Database and Identifier |
|---|---|---|---|---|
| Smithers SP, Shao Y, Altham J, Bex PJ | 2023 | SamSmithers/ Supplementary_materials-Large_differences_in_ target-flanker_depth_ can_increase_crowding: Smithers.eLife2023.v1 | https://doi.org/10. 5281/zenodo.8274202 | Zenodo, 10.5281/ zenodo.8274202 |

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
