## [Editor Report]

Using a novel multi-depth plane display, this important study reveals that crowding decreases with small depth differences between the target and flankers but increases with larger depth differences. The evidence supporting the claims is convincing, although the explanation of some findings is somewhat speculative. This paper will be of interest to visual scientists and neuroscientists.

---

## [Decision Letter]

**Decision letter after peer review:**

Thank you for submitting your article "Large differences in target-flanker depth increase crowding- evidence from a multi-depth plane display" for consideration by *eLife*. Your article has been reviewed by 3 peer reviewers, including Marisa Carrasco as the Reviewing Editor and Reviewer #1, and the evaluation has been overseen by Chris Baker as the Senior Editor.

Essential revisions

1) In general, there are some interactions that were reported, and others that were not reported, but it would be important to know if they are significant (pages 15-16). For example, when the target is at fixation and the target is at a variable flanker depth: In Experiment 1, was there a significant interaction between (a) target-fixation depth and flanker depth (in front versus behind) and (b) target-fixation depth and target-flanker spacing? In Experiment 3, it is reported that perceptual error was higher when the target was in from or behind the flanker ring and fixation and that the greatest perceptual error occurred when the target was behind, but it is not reported if this interaction was significant. Its presence if important to know whether the data should be independently analyzed for 'in front' and 'behind'. In Experiment 5, was the interaction between target-flanker spacing and depth significant?

2) The findings are clear but the explanation(s) for the findings is not. The authors state that large interocular disparity differences likely induce diplopia, which could increase perceptual error by increasing the number of features. The authors should explain what they mean by features and how an increased perceived number of features would increase crowding. Moreover, the authors acknowledge that only a few observers reported experiencing diplopia; however, they speculate that observers may have experienced diplopia but not noticed it consciously given the short stimulus presentation time.

3) At several points in the paper authors refer to the 'natural three-dimensional scenes'. Indeed, the authors increase the ecological validity of their experiment by introducing actual depth differences, therefore allowing for depth cues such as accommodation, vergence and defocus blur. This is indeed a significant improvement over previous studies. However, they still use relatively impoverished visual stimuli in a tightly controlled psychophysical experiment requiring head stabilization by means of a chin rest. So, their experiment is still far removed from deploying actual, ecologically valid, conditions. Consequently, their stimuli mostly lack the complexity and associated clutter of natural stimuli as well as other potential depth cues that an observer might gain from parallax, aerial perspective, lighting, or shading. Therefore, their suggestion "that crowding has a more significant impact on our perception of natural three-dimensional environments than previously estimated with 2D displays." is stretching what can be concluded from their present work. Please temper these comments.

4) The inclusion of a large number of participants, in which none of the participants seemed to have performed all the conditions, is both a strength and a potential weakness. Their current approach of including (presumably) naive participants and having each do a portion of the experiments in itself is valid. But it also adds to the complexity of their study and presumably adds variability to their data.

Including data from a few well-trained participants (some of which could be the authors) performing all the conditions would significantly strengthen the manuscript.

*Reviewer #1 (Recommendations for the authors):*

– (lines 205-206) any reason why fixation time differed between Experiments 1, 3, and 5 and Experiments 2 and 4?

– (line 217) the authors state that participants underwent training, but they do not specify either how many trials or whether there was a criterion in performance that observers should reach before starting the experiments.

– It would be useful if the axes in the Supplementary figure were the same as in the related main figures.

– The authors may want to discuss whether they think their findings would hold at different eccentricities.

*Reviewer #3 (Recommendations for the authors):*

1. The authors measure but do not introduce nor discuss well why they analyze the "proportion of trials in which observers reported seeing the target inside the flanker ring" as an additional dependent variable. This only becomes somewhat clear in the discussion. Here, the usage of this variable as a control variable to discuss the effect of target misalignment on perceptual error becomes clear and useful to interpret the results. However, a much better explanation is needed for why this variable is analyzed as a dependent variable and how this analysis informs about the difference in target-flanker depth on crowding. One may even wonder why the corrected analysis is not treated as the main analysis (as there are good reasons for this control), and the analysis on the unfiltered data in the SI.

2. Following up from weakness 1, the authors should discuss in more detail how these real-world aspects and cues may affect crowding introduced by objects presented at different depth differences. Moreover, they should discuss how this would affect their suggested implication "that crowding has a more significant impact on our perception of natural three-dimensional environments than previously estimated with 2D displays." (citation of abstract).

3. Following up on weakness 2, having a few well-trained participants (some of which could be the authors) perform all the conditions would be a nice addition to the present work. Obviously, in particular, if they show the same pattern of results.

4. I am sure the authors are aware that in crowding research, it is quite common to present results in terms of "crowding magnitude", by subtracting the non-flanker (infinity) condition from the flanker conditions. While this is not a necessity, and their statistics are appropriate, it presumably could remove some of the individual variability in the data and reduce the number of conditions of interest that have to be displayed. At present, this number is quite overwhelming, resulting in somewhat "crowded" figures (pun intended).

5. The sentence in Lines 17 to 19 – "Contrary to previous work showing small differences in target-flanker depth reduce crowding, the larger depth differences tested in our study instead increased crowding" summarizes the result of the study in the abstract. However, instead of highlighting their own result, it tends to emphasize previous work. Therefore, to emphasize their own contribution, I suggest reordering the sentence to something like: "Our study showed that larger differences in target-flanker depth increase crowding, contrary to previous work showing reduced crowding in the presence of small depth differences."

6. Line 68 effect – affect.

7. The text describing the perceived location of the target in Figure 3 step 5 is not readable. The icons by themselves are enough to get the message across. I suggest removing the text (in the picture) from the figure.

8. Increased crowding has also been found in glaucoma: https://doi.org/10.1167/iovs.63.1.36 and https://doi.org/10.1167/iovs.18-25150

9. In the introduction (line 39), in my view, some pioneering modeling work of van den Berg is missing from the citations on "unifying theories". E.g. https://doi.org/10.1371/journal.pcbi.1000646 and https://doi.org/10.1167/12.6.13

10. While explained in the methods section, for readability, I suggest explaining the concept of disparities and diopters in relation to depth perception at an earlier stage.

11. It is not clearly described in the methods that the participants viewed the displays binocularly. Only binocular calibration is mentioned.

12. Line 63 "This challenges the assumption that crowding is a significant problem in three-dimensional natural environments." It is not explained who made this assumption or what it is based on.

13. Line 66 – "that" missing.

14. Line 146 "determined by the Miles test". Suggest briefly describing this test and/or providing a reference.

15. Figure 2 caption: The stimuli was – The stimuli were, or the stimulus was.

16. Line 158 dimeter – diameter.

[Editors' note: further revisions were suggested prior to acceptance, as described below.]

Thank you for resubmitting your work entitled "Large differences in target-flanker depth increase crowding – evidence from a multi-depth plane display" for further consideration by *eLife*. Your revised article has been evaluated by Tirin Moore (Senior Editor) and a Reviewing Editor.

We are glad to see that the manuscript has been improved, as you have satisfactorily addressed most issues raised in the first round of reviews. However, there is a remaining issue that needs to be addressed, explained below by Reviewer #3. The reviewer also recommends how to address this issue. It would strengthen your manuscript to do so.

*Reviewer #3 (Recommendations for the authors):*

The authors have addressed most of my comments satisfactorily.

The authors also well explain their position on why a combination of (condition wise) incomplete data from various groups of participants is still valuable. However, the reason to ask for data in a small group with experienced participants wasn't about not believing the statistics on their large observer pool. It was primarily about whether the reported findings have real-world relevance at the individual level. This may not have been so clear in the first iteration. At the moment, that is hard to conclude, in my view. Indeed, there are statistically significant effects that emerge from the variable data. But is the variance in the data due to the effects themselves being small and highly variable, or is it because the naive participants were simply not very good/stable at indicating their perception, despite the training? In my view, this issue could be resolved by having well-trained, experienced observers repeat the task. If the authors believe they may be biased themselves, they could train a few other observers. I am open to other means to resolve this issue. But it is an actual question/issue that I have/see with the current results.

---

## [Author Response]

Essential revisions1) In general, there are some interactions that were reported, and others that were not reported, but it would be important to know if they are significant (pages 15-16). For example, when the target is at fixation and the target is at a variable flanker depth: In Experiment 1, was there a significant interaction between (a) target-fixation depth and flanker depth (in front versus behind) and (b) target-fixation depth and target-flanker spacing? In Experiment 3, it is reported that perceptual error was higher when the target was in from or behind the flanker ring and fixation and that the greatest perceptual error occurred when the target was behind, but it is not reported if this interaction was significant. Its presence if important to know whether the data should be independently analyzed for 'in front' and 'behind'. In Experiment 5, was the interaction between target-flanker spacing and depth significant?

We apologize for any confusion caused by our reporting of the statistical analysis. As described in the statistical analysis section of the methods, for each experiment we first used model simplification, whereby models were compared with one another using a likelihood ratio test (LRT), to remove non-significant interactions sequentially. This enabled us to test the effect of individual fixed effects that were not involved in an interaction. We then used the minimum model (i.e., model containing all fixed effects of interest plus any significant two-way interactions) to report the significance of each main effect or significant interaction(s) by using a LRT to compare the minimum model with the same model but with the effect/interaction of interest removed. Therefore, in the original version of the manuscript we only reported the results of significant interactions as non-significant interaction were not present in the minimum model. Since, as the review points out, some readers may be interested in the results of these non-significant interactions, we have updated the Results section and now report the results of all non-significant interactions (based on the results of the LRT during model simplification).

2) The findings are clear but the explanation(s) for the findings is not. The authors state that large interocular disparity differences likely induce diplopia, which could increase perceptual error by increasing the number of features. The authors should explain what they mean by features and how an increased perceived number of features would increase crowding. Moreover, the authors acknowledge that only a few observers reported experiencing diplopia; however, they speculate that observers may have experienced diplopia but not noticed it consciously given the short stimulus presentation time.

We concede that the original explanation(s) of our finding in the Discussion section could be made clearer and explained with more detail. Based on these comments we have reworded and expanded the paragraph where we discuss how diplopia, in theory, could increase perceptual error (lines 406-428). This includes adding additional references. We have also made improvements to the abstract (line 17-24) and last paragraph of the discussion (lines 455-470) to make them less speculative.

3) At several points in the paper authors refer to the 'natural three-dimensional scenes'. Indeed, the authors increase the ecological validity of their experiment by introducing actual depth differences, therefore allowing for depth cues such as accommodation, vergence and defocus blur. This is indeed a significant improvement over previous studies. However, they still use relatively impoverished visual stimuli in a tightly controlled psychophysical experiment requiring head stabilization by means of a chin rest. So, their experiment is still far removed from deploying actual, ecologically valid, conditions. Consequently, their stimuli mostly lack the complexity and associated clutter of natural stimuli as well as other potential depth cues that an observer might gain from parallax, aerial perspective, lighting, or shading. Therefore, their suggestion "that crowding has a more significant impact on our perception of natural three-dimensional environments than previously estimated with 2D displays." is stretching what can be concluded from their present work. Please temper these comments.

This assessment of the ecological validity is fair. We agree that while, as the reviewer notes, our study is certainly a significant step forward in our use of large real depth differences, there is still a long way to go in our understanding of how crowding affects perception in complex, real world scenes. Our hope is that this study will help to form the basis and catalysis for this future research. In response to these comments, we have modified the abstract (line 17-24) and discussion (line 455-470) to temper claims concerning ecological validity. We have also added a new figure (Figure 1) to the introduction in response to another reviewer comment which helps to link the motivation for this research to the real world.

4) The inclusion of a large number of participants, in which none of the participants seemed to have performed all the conditions, is both a strength and a potential weakness. Their current approach of including (presumably) naive participants and having each do a portion of the experiments in itself is valid. But it also adds to the complexity of their study and presumably adds variability to their data.Including data from a few well-trained participants (some of which could be the authors) performing all the conditions would significantly strengthen the manuscript.

As stated at the start of the ‘Subject recruitment’ section of the methods, all subjects were naïve and were recruited from the Northeastern University undergraduate population. We agree that it would have strengthened the study if each participant had performed all the conditions, however, each participant was limited to one hour in exchange for course credit as compensation for their time. This is why each participant only performed one of the five experiments.

While the fact each participant did not perform all the conditions may have added variation to the data, we compensated for this by our inclusion of large sample sizes and employed appropriate statistical analysis. This design avoided effects of fatigue or learning that could confound data collected on the same subjects over multiple sessions. Furthermore, we argue that conclusions based on a large sample of naïve participants are more representative of the population than conclusions based on data collected on only a few experienced and potentially non-naïve participants. This is particularly true in cases where some, or all, of the experienced participants are the authors (which is common in human psychophysics) because, depending on the nature of the study, it may be impossible to avoid unconscious bias. While the additional variation in the data from naïve participants does add to the complexity of interpretation, it is nevertheless important to capture and report this variation as it is representative of the natural variation present within the real population.

Given that our results show convincing and statistically supported evidence that large differences in target-flanker depth increase perceptual errors in a naive population, we are surprised that a reviewer would request a replication in a small sample size of experienced, and potentially non-naïve, participants. Aside from being questionable scientifically, this would add redundancy and complexity to an already long manuscript.

Reviewer #1 (Recommendations for the authors):– (lines 205-206) any reason why fixation time differed between Experiments 1, 3, and 5 and Experiments 2 and 4?

The fixation point was displayed for slightly longer before stimulus onset in Experiments 2 and 4 to ensure that observers had adequate time to refocus on the fixation point whenever it switched between the near and far screens. This explanation has been added to the method (lines 561-563).

– (line 217) the authors state that participants underwent training, but they do not specify either how many trials or whether there was a criterion in performance that observers should reach before starting the experiments.

Participants completed as many practice trials as they needed to familiarize themselves with the task. As such the number of practice trials differed between participants. In all cases, the main experiments only began once the participant, and the researcher running the experiment, were confident that they understood the task and were able to perform it correctly, based on the observed responses during practice trials. We have added a sentence to the methods to provide more detail on the training (line 581-584).

– It would be useful if the axes in the Supplementary figure were the same as in the related main figures.

We have changed the Y axis of S1-S4 to be “proportion of trials” (in line with the related figure in the main text) instead of “Number of trials”. We agree that this consistency between supplementary and main figures is better.

– The authors may want to discuss whether they think their findings would hold at different eccentricities.

We have added a sentence to address this on line 459-462 of the discussion.

Reviewer #3 (Recommendations for the authors):1. The authors measure but do not introduce nor discuss well why they analyze the "proportion of trials in which observers reported seeing the target inside the flanker ring" as an additional dependent variable. This only becomes somewhat clear in the discussion. Here, the usage of this variable as a control variable to discuss the effect of target misalignment on perceptual error becomes clear and useful to interpret the results. However, a much better explanation is needed for why this variable is analyzed as a dependent variable and how this analysis informs about the difference in target-flanker depth on crowding. One may even wonder why the corrected analysis is not treated as the main analysis (as there are good reasons for this control), and the analysis on the unfiltered data in the SI.

We agree that the original manuscript didn’t do a good enough job to setup and explain the reason for collecting this variable and performing this analysis. We now introduce and explain this at the start of the Results section (lines 141-147) and have added additional details to the methods (lines 622-628) and link this to the discussion. We chose to include the overall results for perceptual error based on the full data set in the main manuscript as this allow readers to see the results with minimal data manipulation. We also feel that this provides a much better flow to the story being told and sets up an interesting discussion, particularly after the improvements made in response to the reviewers’ comments.

2. Following up from weakness 1, the authors should discuss in more detail how these real-world aspects and cues may affect crowding introduced by objects presented at different depth differences. Moreover, they should discuss how this would affect their suggested implication "that crowding has a more significant impact on our perception of natural three-dimensional environments than previously estimated with 2D displays." (citation of abstract).

Please refer to our response to point 3 in the editor summary section where we discuss the edits made to the discussion concerning the ecological validity and real world implications of our findings. Importantly, the line from the abstract that is quoted here has now been removed from the abstract in response to one of the reviewer’s other comments.

3. Following up on weakness 2, having a few well-trained participants (some of which could be the authors) perform all the conditions would be a nice addition to the present work. Obviously, in particular, if they show the same pattern of results.

Please refer to our response to point 4 in the editor summary section.

4. I am sure the authors are aware that in crowding research, it is quite common to present results in terms of "crowding magnitude", by subtracting the non-flanker (infinity) condition from the flanker conditions. While this is not a necessity, and their statistics are appropriate, it presumably could remove some of the individual variability in the data and reduce the number of conditions of interest that have to be displayed. At present, this number is quite overwhelming, resulting in somewhat "crowded" figures (pun intended).

We agree with the reviewer that "crowding magnitude" is a common way of presenting results within the crowding literature. However, as mentioned within our response to one of the previous comments, we wanted readers to see the results with minimal data manipulation and without individual differences in baseline being lost as a result of this transformation. While, as the reviewer points out, the variation in response does add to the complexity of interpretation, it is still nevertheless important to capture (and visualize) this variation as it is representative of the natural variation present within the population. An appreciation of this natural variation is particularly important for future studies that may seek to replicate some, or all, of the findings of our study.

5. The sentence in Lines 17 to 19 – "Contrary to previous work showing small differences in target-flanker depth reduce crowding, the larger depth differences tested in our study instead increased crowding" summarizes the result of the study in the abstract. However, instead of highlighting their own result, it tends to emphasize previous work. Therefore, to emphasize their own contribution, I suggest reordering the sentence to something like: "Our study showed that larger differences in target-flanker depth increase crowding, contrary to previous work showing reduced crowding in the presence of small depth differences."

We appreciate the reviewer pointing this out. Thank you. We have made the suggested change (with a few edits) to the sentence and agree that it is much better (lines 17-19).

6. Line 68 effect – affect.

Corrected.

7. The text describing the perceived location of the target in Figure 3 step 5 is not readable. The icons by themselves are enough to get the message across. I suggest removing the text (in the picture) from the figure.

Thanks for the suggestion. We have removed the text from the picture for step 5 in what was figure 3 (now figure 4) and instead included it in the figure legend.

8. Increased crowding has also been found in glaucoma: https://doi.org/10.1167/iovs.63.1.36 and https://doi.org/10.1167/iovs.18-25150

Thanks, we have added both references to the introduction (lines 38).

9. In the introduction (line 39), in my view, some pioneering modeling work of van den Berg is missing from the citations on "unifying theories". E.g. https://doi.org/10.1371/journal.pcbi.1000646 and https://doi.org/10.1167/12.6.13

Thanks you, we have added both references to the introduction (line 43) and discussion (lines 339-340 and 403).

10. While explained in the methods section, for readability, I suggest explaining the concept of disparities and diopters in relation to depth perception at an earlier stage.

We now define what a diopter is when it is first mentioned in the introduction (line 92).

11. It is not clearly described in the methods that the participants viewed the displays binocularly. Only binocular calibration is mentioned.

We now state in the methods that the main experiment was conducted under binocular viewing conditions (lines 516-517). This is also stated in the discussion (lines 434-435)

12. Line 63 "This challenges the assumption that crowding is a significant problem in three-dimensional natural environments." It is not explained who made this assumption or what it is based on.

We have adjusted the wording of this sentence and provided a reference (line 66-69) and feel it is greatly improved as a result. We have also added a new figure (Figure 1) to the introduction which helps to further set up the motivation for this research and relate it to the real world.

13. Line 66 – "that" missing.

Corrected.

14. Line 146 "determined by the Miles test". Suggest briefly describing this test and/or providing a reference.

We have added reference for this test (line 511).

15. Figure 2 caption: The stimuli was – The stimuli were, or the stimulus was.

This typo has been corrected to “The stimuli were viewed”.

16. Line 158 dimeter – diameter.

Corrected.

[Editors' note: further revisions were suggested prior to acceptance, as described below.]

We are glad to see that the manuscript has been improved, as you have satisfactorily addressed most issues raised in the first round of reviews. However, there is a remaining issue that needs to be addressed, explained below by Reviewer #3. The reviewer also recommends how to address this issue. It would strengthen your manuscript to do so.Reviewer #3 (Recommendations for the authors):The authors have addressed most of my comments satisfactorily.The authors also well explain their position on why a combination of (condition wise) incomplete data from various groups of participants is still valuable. However, the reason to ask for data in a small group with experienced participants wasn't about not believing the statistics on their large observer pool. It was primarily about whether the reported findings have real-world relevance at the individual level. This may not have been so clear in the first iteration. At the moment, that is hard to conclude, in my view. Indeed, there are statistically significant effects that emerge from the variable data. But is the variance in the data due to the effects themselves being small and highly variable, or is it because the naive participants were simply not very good/stable at indicating their perception, despite the training? In my view, this issue could be resolved by having well-trained, experienced observers repeat the task. If the authors believe they may be biased themselves, they could train a few other observers. I am open to other means to resolve this issue. But it is an actual question/issue that I have/see with the current results.

We thank the reviewer for the clarification, and we can now understand the reasoning behind their recommendation much better. We concede that the question posed by the reviewer, whether the variance in the data is due to the effects themselves being highly variable, or because the naive participants were simply not stable at indicating their perception, is a valid and important issue with the previous versions of the manuscript.

Therefore, as recommended by the reviewer we have repeated the study with a sample of four highly trained and very experienced subjects who participated in all five experiments. As described in the updated methods (lines 594-602, and throughout the methods) all of the experienced subjects (none of whom were the authors) were volunteers from within the university vision research community and were therefore extremely familiar and well-practiced in participating in psychophysics experiments such as those in our current study. The Results section has been updated to include the results from the experiments with these experienced subjects (lines 307-390). We show that the variation in the data from the main experiments is indeed representative of the natural variation between some individuals within the population, and not because the naive subjects were not good/stable at reporting their responses. We have updated the discussion accordingly (lines 400-412, 515-525, and 576-577) and believe it is much stronger as a result. We have also updated the abstract and title to take into account that overall trends at the population level may not always be apparent at the individual level due to natural variation. We are confident that this repeated study with experienced subjects addresses the forementioned question raised by the reviewer and has improved the manuscript.

Additional improvements

We have improved the description of the projector set up by adding additional details (lines 613-614, and 619-626).

Figures 5, 6, 8 and 9, and Figure 5—figure supplement 1 and Figure 6—figure supplement 1 have been updated to make it easier to interpret the second boxplot (A has been made narrower while B has been widened).

Corrected an error in figures 6B and 9B, and Figure 9—figure supplement 2 that meant the results for the no flanker condition were pooled together instead of being plotted separately for the two fixation depths. Also updated Figure 9—figure supplement 1 so it’s formatting is consistent with Figure 9—figure supplement 2.

Correction to Figure 7 which was missing a label for ‘no flankers’ on the x axis.